# The 20S as a stand-alone proteasome in cells can degrade the ubiquitin tag

Indrajit Sahu [1], Sachitanand M. Mali[2], Prasad Sulkshane[1], Cong Xu[3], Andrey Rozenberg[1], Roni Morag[1], Manisha Priyadarsini Sahoo[1], Sumeet K. Singh[2], Zhanyu Ding[3], Yifan Wang [3], Sharleen Day[4], Yao Cong [3,5], Oded Kleifeld [1✉], Ashraf Brik [2✉] & Michael H. Glickman [1✉]

The proteasome, the primary protease for ubiquitin-dependent proteolysis in eukaryotes, is usually found as a mixture of 30S, 26S, and 20S complexes. These complexes have common catalytic sites, which makes it challenging to determine their distinctive roles in intracellular proteolysis. Here, we chemically synthesize a panel of homogenous ubiquitinated proteins, and use them to compare 20S and 26S proteasomes with respect to substrate selection and peptide-product generation. We show that 20S proteasomes can degrade the ubiquitin tag along with the conjugated substrate. Ubiquitin remnants on branched peptide products identified by LC-MS/MS, and flexibility in the 20S gate observed by cryo-EM, reflect the ability of the 20S proteasome to proteolyze an isopeptide-linked ubiquitin-conjugate. Peptidomics identifies proteasome-trapped ubiquitin-derived peptides and peptides of potential 20S substrates in Hi20S cells, hypoxic cells, and human failing-heart. Moreover, elevated levels of 20S proteasomes appear to contribute to cell survival under stress associated with damaged proteins.

[1] Faculty of Biology, Technion–Israel Institute of Technology, Haifa 32000, Israel. [2] Schulich faculty of Chemistry, Technion–Israel Institute of Technology, Haifa 32000, Israel. [3] State Key Laboratory of Molecular Biology, National Center for Protein Science Shanghai, Shanghai Institute of Biochemistry and Cell Biology, Center for Excellence in Molecular Cell Science, Chinese Academy of Sciences, Shanghai 200031, China. [4] Department of Medicine, Perelman School of Medicine, University of Pennsylvania, Philadelphia, PA 19104, USA. [5] Shanghai Science Research Center, Chinese Academy of Sciences, Shanghai 201210, China. ✉email: okleifeld@technion.ac.il; abrik@technion.ac.il; glickman@technion.ac.il

All eukaryotic cells recycle proteins (functional as well as damaged) continuously for survival. The ubiquitin proteasome system (UPS) is the major proteolytic pathway for the removal of cytosolic, nuclear, and many membrane-associated proteins[1,2]. Target proteins are selected by tagging them covalently with ubiquitin, typically with lysine48-linked tetraubiquitin chains, followed by proteolysis within the 26S proteasome[3–5]. The 26S proteasome holoenzyme consists of a 19S regulatory particle (RP; a.k.a. proteasome activator 700), which is responsible for recognizing the ubiquitin signal and unfolding the target protein, and a 20S core particle (CP), which hydrolyzes the unfolded polypeptide into short peptides of varying lengths[6–9]. To do so, the 19S RP utilizes three ubiquitin receptors (PSMD2/Rpn1, PSMD4/Rpn10, and ADRM1/Rpn13), several deubiquitinases that remove the ubiquitin signal, and a hexameric ring of AAA ATPases (RPT1-6 subunits) that unfolds the substrate and translocates it to the barrel-shaped ($\alpha_7\beta_7\beta_7\alpha_7$) 20S CP[10]. After traversing the channel of the ATPase ring, unfolded substrates pass through the $\alpha$-ring at the outer surface of the 20S CP, enter the antechamber, and cross yet another aperture (the $\beta$-annulus) to access the proteolytic chamber of the 20S CP[11–13]. Three of the seven $\beta$ subunits in each of the two $\beta$-rings that align this proteolytic chamber ($\beta$1, $\beta$2, and $\beta$5) are proteolytically active, and between them hydrolyze most substrate peptide bonds to generate a variety of short peptide products[13–15].

Physiological conditions that require broad changes to the proteome necessitate a greater proteolytic capacity to remove the unnecessary proteins. In addition, common stressors such as oxidation, temperature, ionization, or toxins cause protein damage as well as damage to the ubiquitin–proteasome machinery. Thus, such stressors cause a need for alternative pathways for removing damaged proteins. Interestingly, the 20S CP is relatively resistant to oxidation damage compared to the 26S holoenzyme and persists as a stable complex under many stress conditions[16,17]. Hence, it has been suggested that the 20S complex plays a role under stress conditions that correlates with a greater need for removing damaged/misfolded proteins. It is unclear, however, how this role is carried out, as the 20S complex has no associated ATPase domain or dedicated ubiquitin receptors.

Although the 20S subcomplex is an integral part of the 26S holoenzyme, it is also quite abundant as a free complex in many cell types[18]. Free 20S complexes have been suggested to be proteasome assembly intermediates, 26S breakdown products (due to disassembly), or stand-alone proteolytic enzymes[19–23]. For instance, some archaea and prokaryotes, which lack ubiquitin, have free 20S complexes alongside other ATP-dependent proteases. This suggests that the 20S is a primordial protein-degrading machine, which is possibly aided by loosely associated ATPase activators[24,25]. Some reports suggest that the 20S complex functions independently, even in eukaryotes, by acting directly on disordered or oxidized/damaged proteins[26–30]. In eukaryotic cells, the 20S may be augmented by non-ATPase activators such as 11Sreg/PA28 or PA200 in addition to the more abundant ATPase-containing 19S[18,31]. Attachment of proteasome activators influences substrate selection and may also affect product outcome due to allosteric effects on $\beta$-catalytic active sites[32–35]. Thus, changes in the cellular ratio between the 20S and the 26S proteasomes may be part of an adaptive response to meet cellular needs[26,36–38]. However, without an associated unfoldase activity, in vitro 20S CP proteolyzes only unstructured proteins in a ubiquitin-independent manner[39–41].

Since the two proteasome species (26S and 20S) have the same catalytic active sites, understanding their distinctive roles has been challenging. To this end, we chemically synthesized a panel of polyubiquitinated conjugates that are potential substrates for either enzyme. Since the location of the ubiquitin tag on the substrate could influence its binding efficiency to the 26S proteasomes and possibly even the proteolytic outcome[42], we attached lysine48-linked ubiquitin/ubiquitin chains to a natural (i.e., published) site on a known substrate for ubiquitin-dependent 26S proteasome degradation, cyclin B1. Here, we describe the signature activities of the 20S proteasome that could be used to study its function in vivo. For instance, we find that under hypoxia, the 20S proteasome aids clearance of damaged proteins and improves cell viability. We also find, using proteasome-trapped peptides (PTP), that some of these substrates may be proteolyzed along with the conjugated ubiquitin tag.

## Results

**The 20S proteasome shows signature behavior distinct from that of the 26S proteasome.** Misfolded or inherently disordered proteins are often ubiquitinated in cells, even though in vitro they can be proteolyzed by the 20S proteasome. Thus, it is unclear whether in vivo, disordered proteins are degraded by 26S or 20S proteasomes. To explore this, we designed a panel of model substrates based on human cyclin B1, as this cyclin is potentially a substrate for both 20S and 26S proteasomes. Cyclin B1 generally undergoes rapid degradation by the 26S proteasome upon ubiquitination by activated APC/C complex at the end of M-phase of the cell cycle[43]. Structurally, cyclin B1 has a disordered N-terminal region, which contains a degron for recognition by APC/C and 15 lysine residues, many of which can be modified by ubiquitin[44].

Enzymatic ubiquitination of a substrate often results in heterogeneous modifications with respect to polyubiquitin chain length, linkage-type, and the attachment site to the target. This heterogeneity poses a hurdle to the delineation of substrate preferences of proteasomes. Therefore, we chemically synthesized a panel of well-defined homogenous substrates for both enzymes. First, we used a two-step native chemical ligation method to synthesize an 88 amino acid segment from the N-terminus of cyclin B1 (HA-Cyclin B1-NT), in which all lysine residues were changed to arginine except for lysine64 (Fig. 1a, b (3), Supplementary Methods). We then synthesized MonoUb or DiUb linked via lysine48 that were chemically ligated through an isopeptide bond to lysine64 of HA-Cyclin B1-NT (Fig. 1b (5), Supplementary Methods). TetraUb-Cyclin B1-NT was synthesized similarly by ligating DiUb to DiUb-Cyclin B1 (Fig. 1b (7)). To facilitate the tracking of specific ubiquitin units in the chain, the proximal ubiquitin in all chains was Myc tagged, and the distal ubiquitin of TetraUb was Flag tagged (Fig. 1c). The purity and integrity of the final branched protein conjugates were confirmed by analytical HPLC-MS (Supplementary Methods), SDS-PAGE, and immunoblots (IB) (Fig. 1d). In parallel, 26S proteasomes (lacking transiently associated DUBs) and 20S proteasomes were purified from human erythrocytes (Fig. 1e). The purity, activity, integrity, and composition of these purified proteasomes were verified by native-PAGE, in-gel activity assay, SDS-PAGE (Supplementary Fig. 1a, b), and LC-MS/MS (Supplementary Data 1). Only a trace amount of PA28 (PSME1 and PSME2 subunits) or PA200 (PSME4) (Supplementary Fig. 1c), no detectable p97/VCP (Supplementary Data S1), and virtually no measurable amount of deubiquitinases (Supplementary Fig. 1d) were observed in our preparations. The specific peptidase activity of the purified 20S proteasome was further activated by SDS (Supplementary Fig. 1a), as documented for the latent form of 20S complexes[45].

We found that in vitro, unmodified Cyclin B1-NT was proteolyzed faster by purified 20S proteasomes than by 26S proteasomes (Fig. 2a–c), consistent with a recent report, which

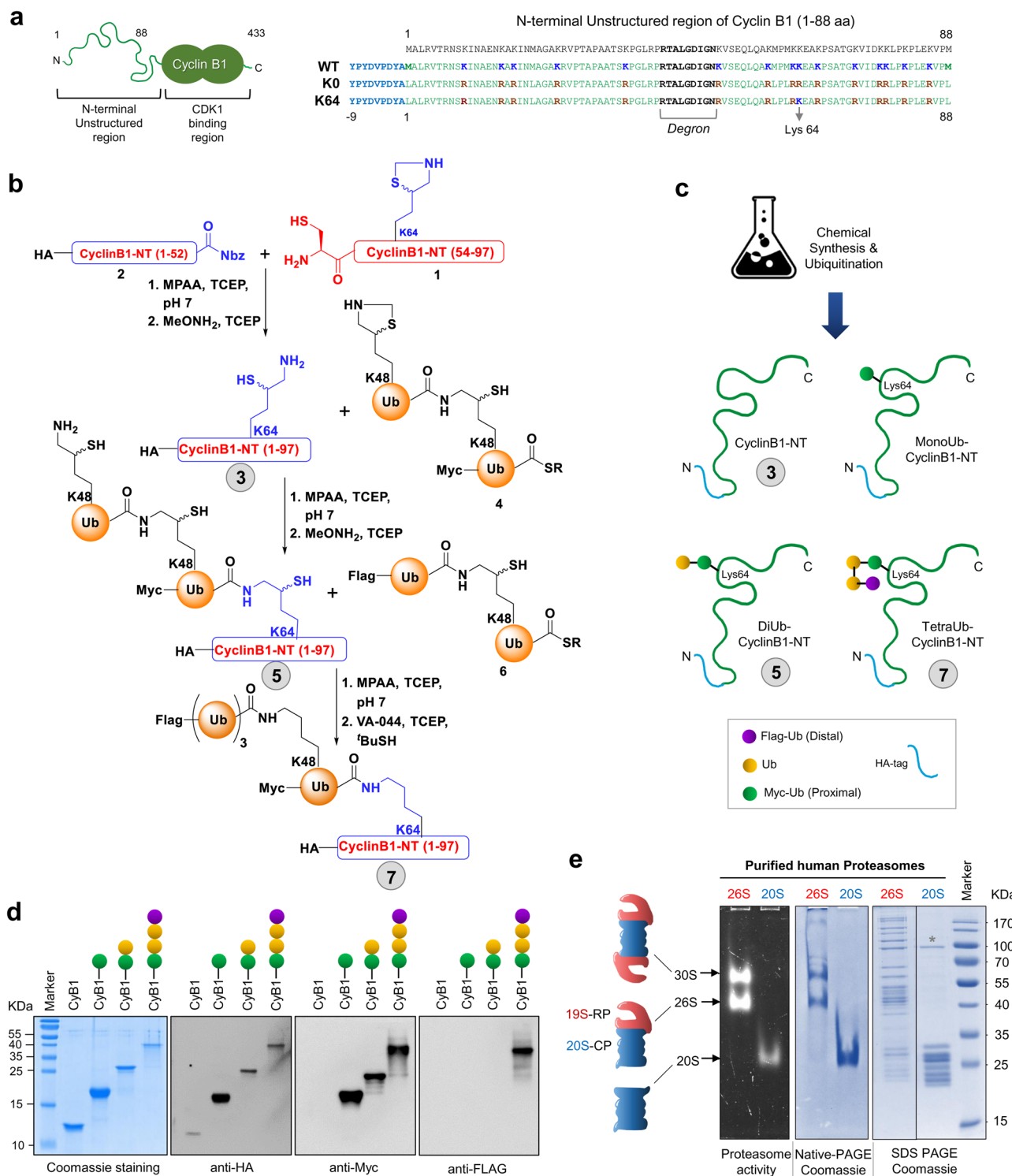

suggested that 20S complexes are more efficient than 26 complexes in degrading a naturally disordered protein[26]. In contrast, TetraUb-Cyclin B1-NT was proteolyzed faster by purified 26S proteasomes, as expected (Fig. 2d, e). A gradual decrease in the rate of Cyclin B1-NT processing by 20S proteasomes was observed, which was proportionate to the number of ubiquitin units attached to the same substrate, whereas the inverse phenomenon was recorded for 26S proteasomes (Fig. 2f, Supplementary Fig. 2a, b). By binding to ubiquitin receptors, ubiquitin units facilitated the degradation of a tagged substrate by the 26S proteasome, which was not the case for the

20S proteasome that lacks ubiquitin receptors. We evaluated whether ubiquitinated proteins are recruited to 20S proteasomes as efficiently as the equivalent unmodified protein, using a competition assay. We compared the proteolysis of Cyclin B1-NT by the 20S proteasome alone and in the presence of equimolar monoUb-Cyclin B1-NT. A marked slowdown of cyclin proteolysis was observed when monoUb-Cyclin B1-NT was added to the reaction mixture (Supplementary Fig. 2c), indicating that the ubiquitinated protein is a competitive substrate for the 20S proteasome and indirectly suggesting that both substrates are comparably recruited to the 20S proteasome. The ubiquitinated

**Fig. 1 Chemical synthesis and ubiquitination of ubiquitin-conjugates. a** A cartoon depicting cyclin B1 with its 1–88 aa residue unstructured N-terminal region (NT), and a C-Terminal CDK1 binding region (left). The primary amino acid sequence of the N-terminal region of cyclin B1 that encompasses the Degron and 15 Lysine residues (right). Below are the wild-type (WT) and K0-mutant (all Lysine to Arginine) Cyclin B1-NT sequences used in this study for mammalian cell expression. The K64 HA-Cyclin B1-NT sequence was used for chemical ubiquitination of substrates (bottom). **b** Schematic illustration of the strategy used to synthesize differentially tagged TetraUbK48-HA-Cyclin B1-NT (Nbz = N-acyl-benzimidazolinones, SR = 3-mercaptopropionic acid). The detailed chemical synthesis protocol is described in Supplementary Methods. **c** A cartoon representation of a set of four synthetic substrates designed to study proteasome function. Tagged (Myc/Flag) and/or untagged ubiquitin units were attached to lysine64 of the HA-Cyclin B1-NT to obtain ubiquitinated conjugates. The numbers 3, 5, and 7 correspond to the three synthetic substrates in panel (**b**) (see also Supplementary Methods). **d** All four synthetic ubiquitinated Cyclin B1-NT conjugates were resolved by SDS-PAGE and stained with Coomassie or probed with the indicated antibodies for immunoblotting (IB). **e** Human 30S/26S and 20S proteasomes purified from human erythrocytes were resolved by native gel for in-gel activity (left), or for Coomassie (center), or resolved by SDS-PAGE for Coomassie staining. The band marked by * near 90 kDa was identified as HSP90AA1 and HSP90AB1 in MS/MS analysis. All other bands were confirmed to contain primarily 20S (right) or 26S (to its left) subunits. Source data are provided as a Source data file.

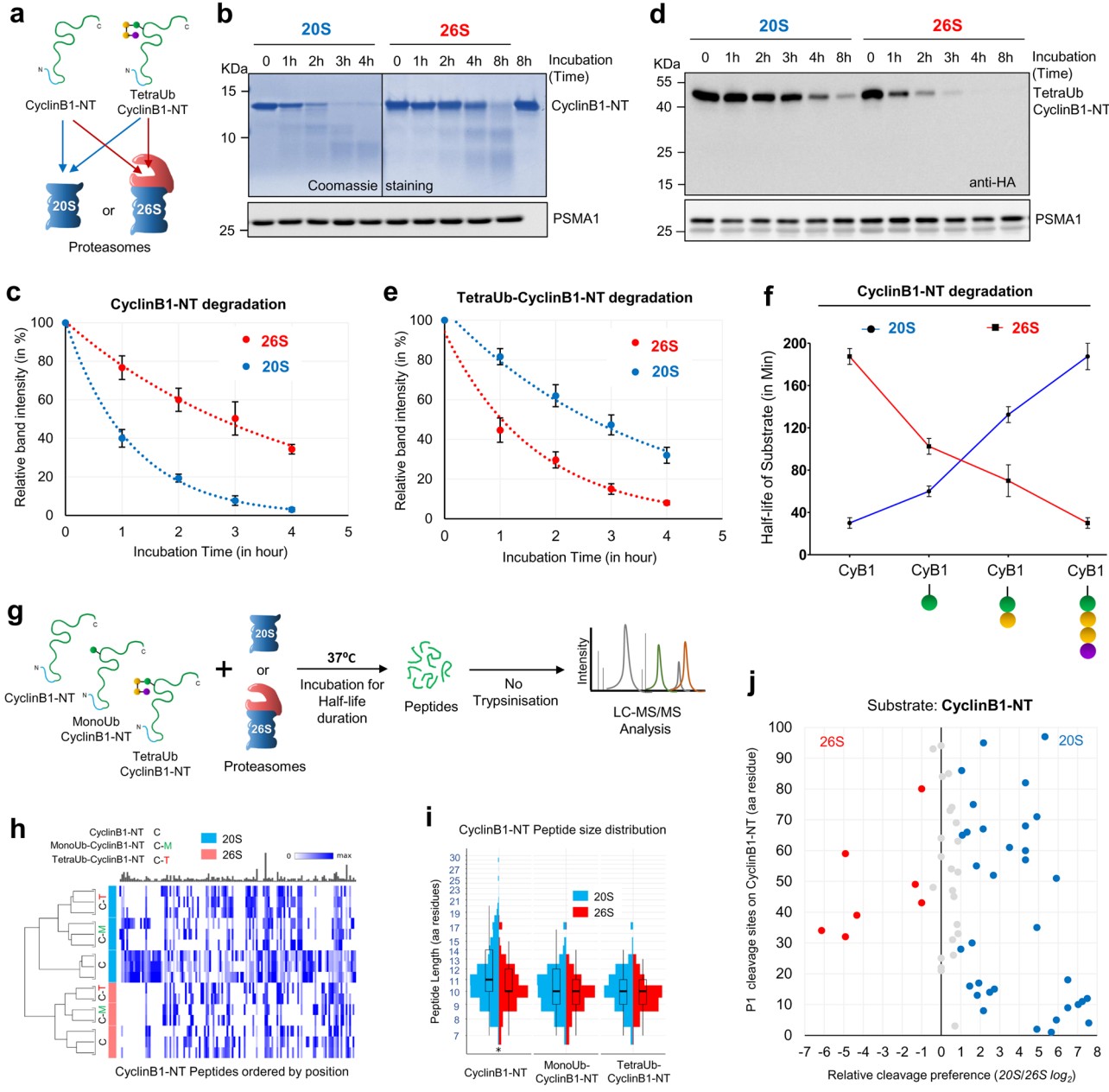

**Fig. 2 The 20S proteasome processes substrates differently than the 26S proteasome. a** Unmodified Cyclin B1-NT or TetraUb-Cyclin B1-NT were incubated at 37 °C with either purified 20S or 26S proteasomes at 1:150 (proteasome:substrate) molar ratio. **b** Unmodified Cyclin B1-NT after reaction with purified 20S or 26S proteasomes for indicated time periods was resolved by 16% Tris-Tricine denaturing gel for Coomassie staining. IB with anti-PSMA1 antibody used as proteasome loading control. **c** The average percentage of residual unmodified Cyclin B1-NT substrate at each time point after degradation by either 20S or 26S proteasomes (from **b**). Error bar represents the value for three independent experiments with ±SD. **d** TetraUb-Cyclin B1-NT was incubated with either purified 20S or 26S proteasomes for indicated time periods. The reaction mixture was resolved by SDS-PAGE for IB with anti-HA (substrate) or anti-PSMA1 (loading control) antibodies. **e** The average percentage of residual TetraUb-Cyclin B1-NT substrate at each time point after degradation by either 20S or 26S proteasomes. Error bar represents the value for three independent experiments with ±SD. **f** Average half-life of differentially ubiquitinated Cyclin B1-NT species in presence of either 20S or 26S proteasomes. Error bar represents the value for three independent experiments with ±SEM. **g** Experimental design to identify products of unmodified Cyclin B1-NT, MonoUb-Cyclin B1-NT, or TetraUb-Cyclin B1-NT incubated with either purified 20S or 26S proteasomes. Peptide products at time points shorter than substrate half-life for each reaction were isolated and analyzed by LC-MS/MS. **h** Heatmap of Cyclin B1-NT-derived peptides obtained from each substrate by 20S or 26S proteasomes (triplicate or duplicate experiments). Each peptide product is positioned based on its N-terminal residue along the primary sequence of Cyclin B1-NT. Color intensity reflects PSM counts for each peptide normalized to the maximum observed counts. The bar plot above (in gray) represents relative sizes of the corresponding maxima. The dendrogram to the left was obtained by performing MDS analysis on the PSM counts of the top 100 peptides and clustering the samples based on the corresponding MDS distances. **i** Size distribution of Cyclin B1-NT peptide products in each sample from panel (**g**). The horizontal box lines represent the first quartile, the median, and the third quartile. Whiskers denote the range of points within the first quartile −1.5× the interquartile range and the third quartile +1.5× the interquartile range. $n = 3$ independent experiments. **j** Scatter plot represents the relative cleavage preference for each proteasome species at P1 positions on unmodified Cyclin B1-NT. Cleavage preference of each enzyme was calculated from the MS/MS count of each peptide product contributing to a given P1 site. The relative ratio plotted as log2 (described in the "Methods" section). The Y-axis represents the amino acid residue number of the HA-Cyclin B1-NT sequence. Red/blue color dots indicate a greater-than-twofold preference for a given P1 cleavage site (red, 26S; blue, 20S; gray, no significant preference). Source data are provided as a Source data file.

substrate associated to the 20S proteasome via the unstructured stretch, but its proteolysis was attenuated due to the globular ubiquitin domain. Overall, rapid proteolysis of a ubiquitinated unstructured protein is a characteristic of the 26S proteasome, whereas rapid proteolysis of a non-ubiquitinated form of the same protein appears to be a signature of the 20S proteasome.

Next, we mapped the degradation products of the 20S and 26S proteasomes. Since the two complexes display preferences for substrates at different ubiquitination states, and they proteolyze these shared substrates at different rates, we were intrigued to analyze their peptide products. To this end, we incubated unmodified Cyclin B1-NT as well as MonoUb- and TetraUb-Cyclin B1-NT with each proteasome and analyzed the degradation products by MS/MS. In order to minimize product reprocessing, great excess of the substrate over the enzyme was maintained and proteolysis was terminated early, so that no more than 40% of the initial substrate was degraded at any time point (Fig. 2g). After quenching the reaction, the peptide products were separated from other proteins by ultra-filtration under denaturating conditions and subjected to proteomic analysis without trypsin digestion. MS/MS analysis of the isolated peptide products showed differential cleavage patterns and peptide distribution for cyclin B1 with or without ubiquitin attached (Fig. 2h, Supplementary Data 2, 3, 4). For all substrates, the 20S proteasome generated longer peptides than the 26S proteasome (Fig. 2i), which is consistent with earlier studies that measured cleavage patterns of non-ubiquitinated loosely folded substrates[32–35,46]. Thus, generating longer peptides is a second signature of the 20S proteasome compared to the 26S proteasome.

Even though 20S and 26S proteasomes did not differ in their overall amino acid preference at the cleavage site (Fig. 2j, Supplementary Fig. 3a, b), different product repertoires by the two enzymes resulted from a marked preference for various P1 sites (Supplementary Fig. 3a). This difference in product repertoires implies that sequence information beyond the adjacent residue contributes to the P1 preferences by each enzyme (e.g., access of the internal peptide bonds to the catalytic β subunits could be influenced by translocation of the substrate through the proteolytic channel). Generally, the diversity of peptide products was lower with the 26S proteasome than with the stand-alone 20S catalytic chamber, or with ubiquitination of

the substrate (Supplementary Fig. 3c), which could reflect greater flexibility of the 20S proteasome to access potential cleavage sites. Thus, a distinct cleavage pattern by the 20S proteasome is a third signature of the 20S compared to the 26S proteasome.

**The 20S proteasome degrades ubiquitin along with an unstructured conjugated substrate**. 26S proteasome-associated DUBs release the ubiquitin tag from the substrate as a mechanism to recycle ubiquitin. As we observed that the 20S proteasome proteolyzed ubiquitinated Cyclin B1-NT in vitro, we next investigated how this complex, without associated deubiquitinase or unfoldase activities, handles the ubiquitin tag. We found that, unlike 26S proteasomes that released tetraubiquitin en-bloc, the 20S proteasome proteolyzed TetraUb-Cyclin B1-NT in its entirety without releasing any free ubiquitin (Fig. 3a, Supplementary Fig. 4a). Likewise, MonoUb- or DiUb-Cyclin B1-NT were also proteolyzed entirely by 20S proteasomes (Fig. 3b). To confirm that this activity was due to 20S proteasomes and to rule out potentially associated proteases, ATPases, deubiquitinases, or chaperones (Hsp90), we repeated the reactions in the presence of relevant inhibitors. None of the tested inhibitors (except for the proteasome inhibitor MG132) blocked proteolysis of MonoUb-Cyclin B1-NT by 20S proteasomes (Supplementary Fig. 4b). Notably, 20S proteasomes showed no detectable deubiquitinase or proteolytic activity on free lysine48-linked polyUb, TetraUb, or DiUb chains, thus ruling out trace amounts of contaminating DUBs (Supplementary Fig. 5a–c). As the 20S proteasome did not proteolyze free unanchored chains, we hypothesized that it degrades ubiquitin only when the ubiquitin is conjugated to an unstructured substrate. The ability to proteolyze a globular domain is particularly surprising given that fused GFP or RFP was reported to be released from unstructured proteins by the 20S proteasome[42,46]. Therefore, we wished to confirm the proteolysis of the conjugated ubiquitin to cyclin B1. To this end, the 20S proteasome reactions were quenched, all peptide products were collected and subjected to MS/MS analysis (Fig. 3c, Supplementary Data 5). We identified peptides spanning the entire poly-ubiquitin chain including the Myc and Flag epitopes on the proximal and distal ubiquitin units, respectively. Chimeric (non-tryptic) peptides spanning these epitopes provided additional

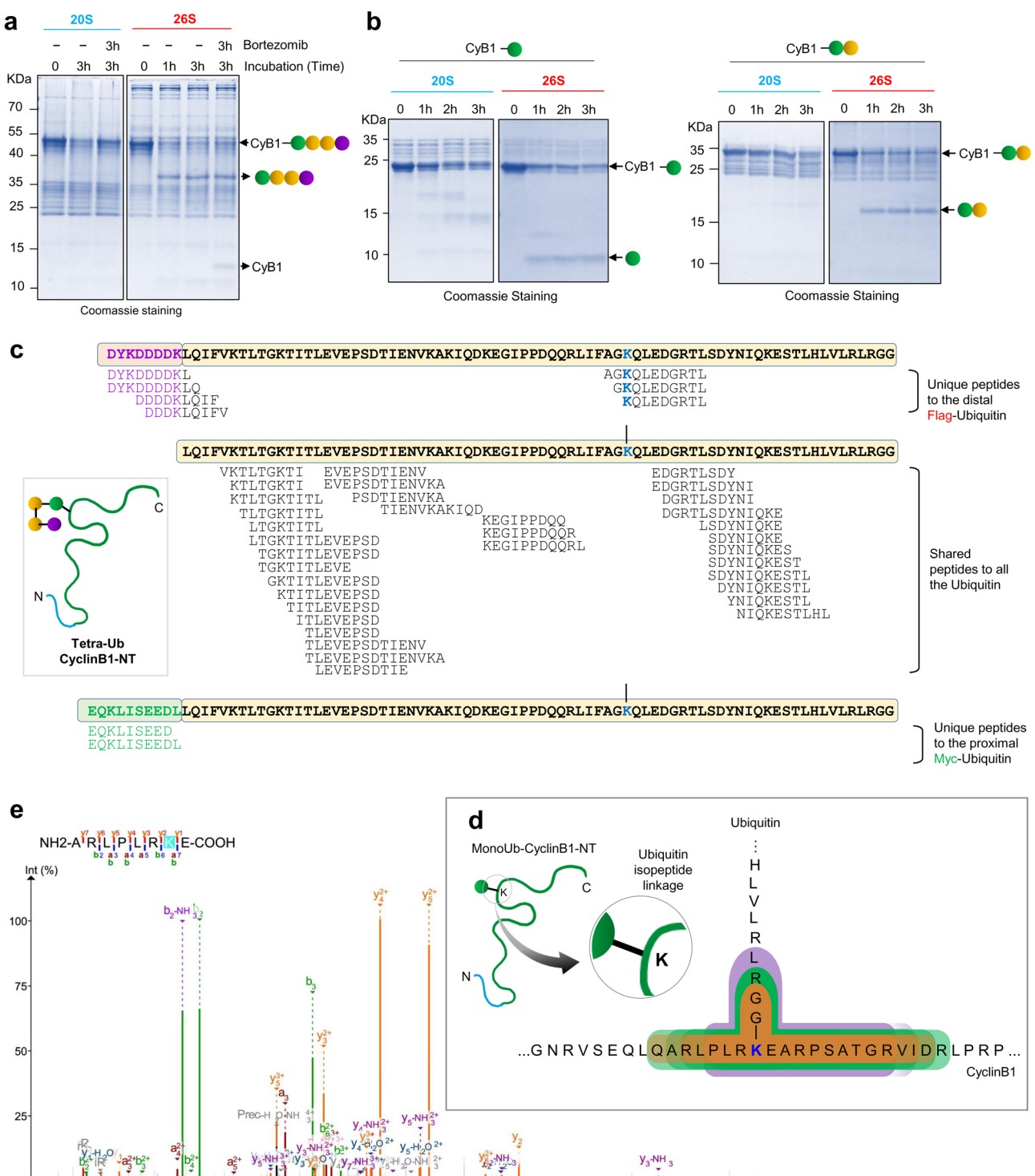

**Fig. 3 Conjugated ubiquitin is proteolyzed by purified 20S proteasome in vitro. a** TetraUb-Cyclin B1-NT was incubated with purified 20S or 26S proteasomes at 1:150 (proteasome:substrate) molar ratio at 37 °C for indicated time periods. The reaction mixture was resolved by Tris-Tricine PAGE and stained with Coomassie. Bortezomib was added to reactions to inhibit the proteasome activity. **b** MonoUb- or DiUb-Cyclin B1-NT were incubated at 37 °C with purified 20S or 26S proteasomes at 1:150 (proteasome:substrate) molar ratio for indicated time periods and separated in Tris-Tricine PAGE followed by Coomassie staining. **c** The illustration represents the ubiquitin peptides obtained from TetraUb-Cyclin B1-NT proteolyzed by the 20S proteasome detected by MS/MS analysis overlaid on the primary sequences of the ubiquitin units in the chain. Purple-colored residues represent the Flag-tag sequence on the distal ubiquitin unit, while green represents the Myc-tag sequence on the proximal ubiquitin. **d** The illustration represents the branched peptides at the isopeptide linkage of the C-terminus of ubiquitin to the K64 residue of cyclin B1 identified by non-tryptic MS/MS of reaction products. Three major types of branched peptides at K64 of Cyclin B1 were detected with variants of ubiquitin remnants: K64—GG, K64—GGR, and K64—GGRL. **e** MS/MS spectra of one of the detected branched peptides ARLPLPKE with a GGR branch on K. Source data are provided as a Source data file.

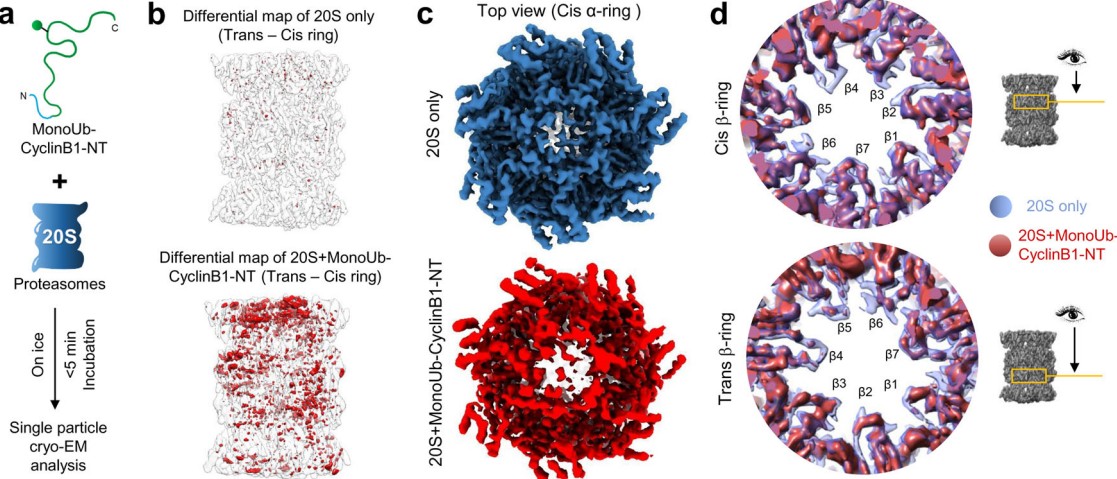

**Fig. 4 Conjugated MonoUb-Cyclin B1-NT induces structural alterations to the 20S proteasome. a** The model depicts the work-flow of sample preparation for cryo-EM: MonoUb-Cyclin B1-NT was incubated briefly with purified 20S proteasomes at 1:5 (proteasome:substrate) molar ratio on ice, frozen within 5 min, and then vitrified for further cryo-EM single-particle analysis. **b** Difference map rendering the symmetric pattern between the *trans*- and *cis*-rings of the 20S-alone map (upper) and 20S + monoUb-Cyclin B1-NT map (lower). The difference map (red density) was generated by using a 180° rotated (around the pseudo twofold axis) map minus its original unrotated map (transparent gray). **c** The top view of density maps from the cis α-ring of 20S only (blue) and the S1 class of 20S + monoUb-CyclinB11-NT (brick red) dataset. **d** Comparison of the 20S-alone (transparent blue) and the S1 class of 20S + MonoUb-Cyclin B1-NT (brick red) maps in the two β-rings focused on the β-annulus regions. Note the density loss in some of the annulus loops or β-sheet regions exposed to or close to the catalytic chamber in both β-rings.

evidence for proteolysis of the entire TetraUb chain by 20S proteasomes (Fig. 3c, Supplementary Data 5).

As ubiquitin-derived peptides were only identified from ubiquitin that was conjugated to a substrate, we posited that the ubiquitin polypeptide accompanies the target substrate into the proteolytic chamber. We addressed the possibility of a branched peptide entering into the proteolytic chamber, by adopting the MSFragger tool to search for variable ubiquitin remnants on proteasome-digested cyclin B1 (Supplementary Data 6). A total of 18 different branched peptide sequences on lysine64 of cyclin B1 were identified with high confidence based on their MS/MS fragmentation spectra (Fig. 3d, e, Supplementary Table 1). This is direct evidence of the ability of the 20S proteasome to process a branched polypeptide, specifically, in this case, to cleave around an isopeptide bond.

In order to better understand how a ubiquitin conjugate enters the proteolytic chamber of the 20S complex, we performed cryo-EM single-particle analysis on purified human 20S proteasomes before and after incubation with MonoUb-Cyclin B1-NT (Fig. 4a). Classification of the data from untreated 20S proteasomes resolved one class of symmetric conformation (further classification did not yield significant differences between classes; Supplementary Fig. 6). A similar analysis of substrate-incubated 20S complexes resulted in five classes, with one class (S1) comprising 29.1% of the particles in an asymmetric appearance (Supplementary Fig. 7a–e). 62.9% of the particles appeared symmetric, closed at both α-rings similar to the 20S only structure, and could be refined into a map denoted as S2 (Supplementary Fig. 7d, g). S2 may represent the particles not directly or obviously affected by substrate. Notably, the angular distribution of the two data sets was similar (Supplementary Fig. 8a). Even after mixing the two data sets, 3D classification yielded similar classes with similar angular distribution (Supplementary Fig. 8b–d). Incubating with a ubiquitinated substrate for a mere 5 min resulted in asymmetric structural alterations to the 20S barrel by destabilizing subunits in the cis α-ring of S1 relative to S2 (Supplementary Fig. 7f, g) and the proteolytic chamber (Fig. 4b). Discontinuous density in the cis α-ring was particularly

notable at the N-termini of α2/α3/α4 subunits that define the gate at the center of the ring (Fig. 4c). Discontinuous density could reflect potential conformational dynamic variations in the gate region of the affected maps over an area of sufficient dimensions to enable entry of the branched polypeptide at the site of ubiquitin conjugation to Cyclin B1-NT. This conformation is reminiscent of the partially open-gate form of activated 20S CP[14]. The above conformational dynamic variations were also apparent after further processing by DeepEMhancer[47]. Although the processed density maps still show distinct variations, models generated from the two maps largely overlap. Careful observation of the models reveals variations in the gate region (Supplementary Fig. 9a) and in particular to the reverse turn of several α-subunits (located between the N-terminal tail and H0 helix; Supplementary Fig. 9b). We note that the reverse turns are key factors in 20S proteasome gate opening[48].

Comparing density maps of 20S complexes with and without substrate also highlighted the flexibility at beta annulus loops in both β-rings (Fig. 4d). 3D variability analyses (3DVA) map using CryoSparc also supported the dynamic nature of the gate region of the 20S complex in the presence of monoUb-Cyclin B1-NT, which might be caused by substrate association (Supplementary Fig. 10). These substrate-induced structural alterations are all consistent with a protein substrate entering through the center of one of the α-rings into the inner proteolytic chamber enclosed by the β-rings. Allosteric effects between the α-gate and β-active sites have been suggested to couple substrate translocation and proteolysis[49]. The surprising ability of the 20S proteasome to engage ubiquitin conjugates and proteolyze ubiquitin along with its attached target protein is the fourth signature of 20S proteasomes, which contrasts with ubiquitin recycling by 26S proteasomes.

**Signature activities of the 20S proteasome recapitulated in cells.** Next, we determined whether the signature properties of purified 20S proteasomes are also detectable in intracellular proteolysis. To this end, we adapted the cyclin B1 model-

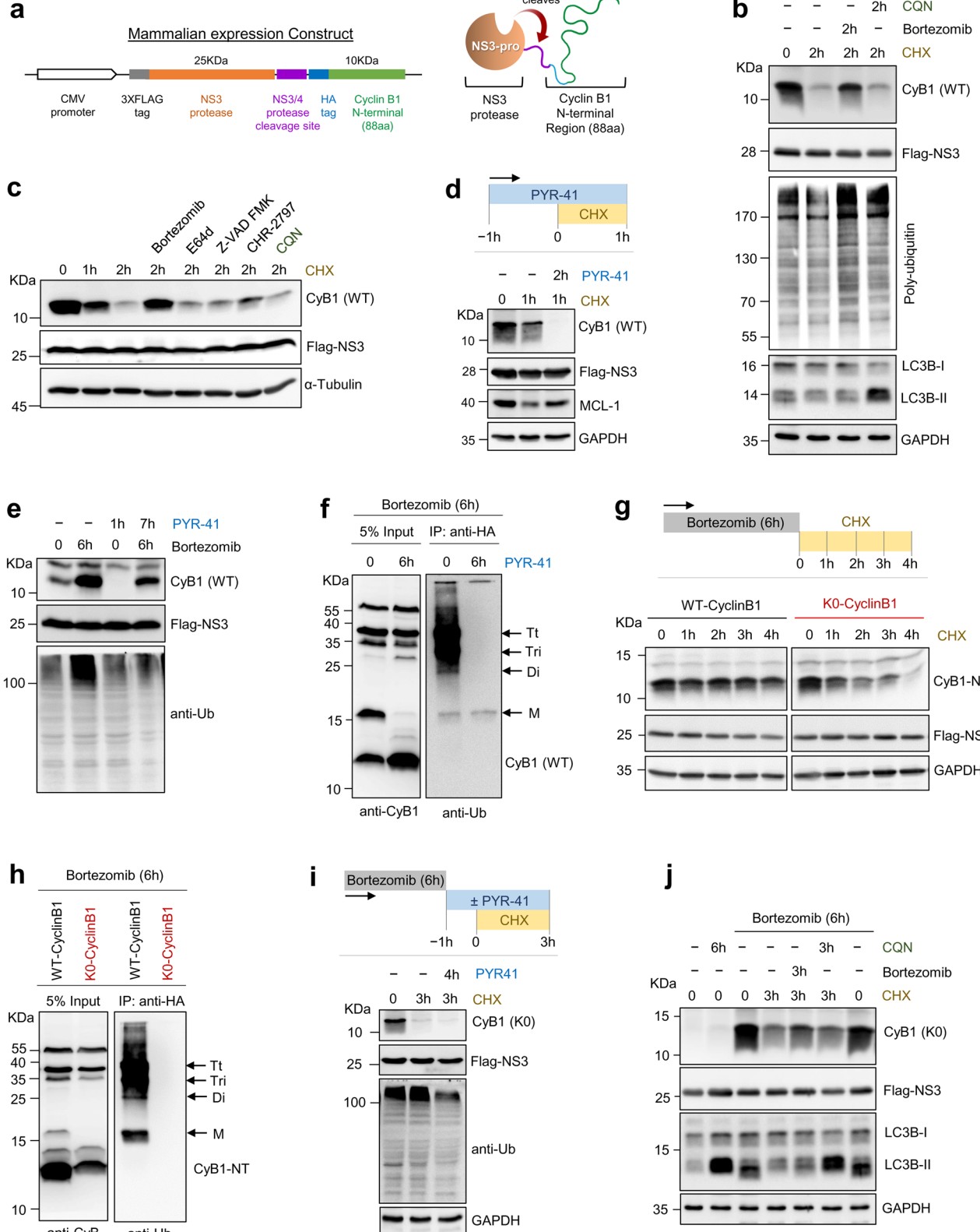

substrate for expression in cell culture by tagging Cyclin B1-NT (1–88 aa) with an HA epitope and fusing it N-terminally to a self-immolating NS3-protease (Fig. 5a). After synthesis, cleavage of the stable NS3 domain served as an expression standard (Supplementary Methods), whereas the unstructured Cyclin B1-NT segment underwent turnover over rapidly. Proteolysis of the Cyclin B1-NT polypeptide in HEK293T cells was sensitive to a

proteasome inhibitor (Fig. 5b) but not to inhibitors of other plausible proteolytic pathways (Fig. 5c). Nevertheless, although typically Cyclin B1-NT is ubiquitinated, surprisingly, Cyclin B1-NT was degraded faster when E1 was blocked, unlike another ubiquitin-dependent substrate, MCL-1[50]. MCL-1 was stabilized in presence of the E1 inhibitor PYR-41, as expected (Fig. 5d and Supplementary Fig. 11a). In fact, ubiquitin-independent

**Fig. 5 Cyclin B1-NT is efficiently proteolyzed in a ubiquitin-independent pathway. a** Cloning strategy for expression of HA-Cyclin B1-NT chimera fused to the NS3-protease in a mammalian expression vector (left). The resulting fusion protein of cyclin B1 N-terminus (residues 1–88 aa) with the NS3-protease (right). **b** HEK293T cells expressing HA-Cyclin B1-NT were treated with cycloheximide (CHX; 50 μg/mL) and/or bortezomib (1 μM) and/or chloroquine (CQN; 50 μM) for 2 h followed by IB for the indicated proteins. **c** HEK293T cells expressing HA-Cyclin B1-NT were treated with cycloheximide (50 μg/mL) and/or bortezomib (1 μM)/E64d (10 μM)/z-VAD-FMK (50 μM)/CHR-2797 (10 μM)/chloroquine (50 μM) for 2 h followed by IB for the indicated proteins. **d** HEK293 cells expressing HA-tagged Cyclin B1-NT were treated with 50 μg/mL cycloheximide for 1 h, with or without E1 inhibitor (PYR-41) at 25 μM and IB for the indicated proteins. **e** HA-Cyclin B1-NT expressing HEK293T cells were treated with/without bortezomib (1 μM) and/or PYR-41 (10 μM) for indicated time periods followed by IB as indicated. **f** HA-Cyclin B1-NT expressing HEK293T cells were treated with bortezomib (1 μM) and/or PYR-41 (10 μM) for 6 h followed by immunoprecipitation of HA-Cyclin B1-NT using anti-HA antibody. IB was performed using anti-ubiquitin antibody for immunoprecipitated fraction and anti-Cyclin B1-NT for 5% Input. **g** HEK293T cells expressing either HA-tagged WT- or K0-Cyclin B1-NT were treated with a pulse of bortezomib at 0.5 μM concentration for 6 h followed by cycloheximide chase. Various proteins in cell lysates were detected by IB. **h** Wild type (WT) or K0-Cyclin B1-NT were trans-expressed in HEK293T cells followed by 6 h pulse treatment of bortezomib (0.5 μM). HA-Cyclin B1-NT was immunoprecipitated using anti-HA antibody. Elution of immunoprecipitated fraction was IB using anti-ubiquitin antibody, 5% of Input was IB using anti-Cyclin B1-NT. **i** K0-Cyclin B1-NT expressing HEK293T cells were pulse treated with bortezomib (0.5 μM) for 6 h. Then bortezomib was removed and cells were grown with/without treatment of cycloheximide (50 μg/mL) and/or PYR-41 (10 μM) for indicated time periods followed by IB as indicated. **j** K0-Cyclin B1-NT expressing HEK293T cells were pulse treated with bortezomib (0.5 μM). Then bortezomib was removed and cells were grown with/without treatment of cycloheximide (50 μg/mL) and/or bortezomib (1 μM)/chloroquine (50 μM) for 2 h followed by IB as indicated. Source data are provided as a Source data file.

proteolysis of Cyclin B1-NT was still dependent on proteasomes and appeared faster than the turnover of ubiquitinated cyclin B1 (Fig. 5e, f). To confirm the ubiquitin-independent nature of Cyclin B1-NT degradation using another approach, we expressed a lysine-less mutant (K0) that cannot be ubiquitinated and found that its cellular half-life was shorter than its wild-type counterpart (Fig. 5g and Supplementary Fig. 11b–d). Possible non-lysine ubiquitination of K0-Cyclin B1-NT was ruled out as no ubiquitination of this species was detected (Fig. 5h). Moreover, degradation of the K0 Cyclin B1-NT construct was also sensitive to proteasome inhibitors (as was the original lysine-containing version; Fig. 5g and Supplementary Fig. 11b), and it was unaffected by E1 or lysosomal inhibitors (Fig. 5i, j). Hence, K0-Cyclin B1-NT was a likely substrate for 20S proteasomes in cells.

In order to evaluate the contribution of 20S proteasomes toward ubiquitin-independent proteolysis, we engineered a HEK293T cell line abundant in 20S complexes by knocking down the gene for the essential 19S RP subunit—PSMD2/Rpn1 (Supplementary Fig. 12a) based on a reported approach[51]. The resulting Hi20S cell line harbored an abnormally high ratio of 20S to 30S/26S proteasomes (>80% of the total; Fig. 6a and Supplementary Fig. 12b). Turnover of K0-Cyclin B1-NT (which cannot be ubiquitinated) in these Hi20S cells was faster than in control cells (Fig. 6b, Supplementary Fig. 12c). In contrast, degradation of WT-Cyclin B1-NT was slower in Hi20S compared to WT cells, confirming that ubiquitination slowed down the degradation of Cyclin B1. In vitro, the non-ubiquitinated form of another loosely folded protein, α-Synuclein[52,53], was reported to be degraded by the 20S proteasome[54]. Indeed, the turnover of unmodified α-synuclein was faster in Hi20S cells than in control cells (Supplementary Fig. 12d). Together, these results not only support a role for the 20S as an active proteasome in cells, but also suggest that for certain unstructured proteins, the 20S proteasome is a faster protease than the 26S proteasome due to the preference of the latter for ubiquitinated substrates.

Having confirmed that cyclin B1 is a likely substrate of 20S proteasomes in cells, we next examined whether the signature features of the 20S proteasome identified in vitro could be recapitulated in cells. To this end, we mapped all the peptides generated from cyclin B1 in cells and compared the entire repertoire to the products generated from cyclin B1 by 20S proteasomes in vitro (Supplementary Fig. 13a, and Supplementary Data 7, 8). We identified a greater variety of cyclin B1-derived peptides in Hi20S than in control cells (Fig. 6c), consistent with the characteristic cleavage patterns of isolated

20S and 26S proteasomes in vitro reactions (Fig. 2h). Moreover, several unique P1 cleavage sites detected in Hi20S cells matched the signature cleavage pattern for purified 20S proteasomes (Fig. 6c). These Hi20S cells also generated longer peptides from K0-Cyclin B1-NT than were generated in control cells (Fig. 6d). Analysis of the total intracellular peptidome also showed a longer peptide size distribution in Hi20S cells compared to control cells (Fig. 6e), consistent with the signature behavior of 20S proteasomes in vitro. In the total pool of intracellular peptides that were captured, 787 target substrate proteins were unique to Hi20S cells (Supplementary Fig. 13b). These substrates had a higher median protein disorder score than those from control cells (Fig. 6f), consistent with disordered proteins targeted preferentially to 20S proteasomes over 26S proteasomes.

**Evidence for a possible contribution of 20S proteasomes to intracellular proteolysis.** Having noticed that the intracellular peptidome in Hi20S cells reflected proteins with a higher disordered score than the peptidome of control cells, we posited that this reflects a preference of 20S proteasomes for proteolysis of substrates with disordered segments. Therefore, we evaluated the effect of changing 20S proteasome levels on the removal of disordered proteins. To do so, we induced misfolded/damaged proteins in cells, documented their ubiquitination status, and evaluated their removal. Control and Hi20S cells were treated with puromycin in order to generate newly synthesized misfolded polypeptides and follow their clearance. Puromycin specifically replaces an aminoacylated tRNA at the ribosome and incorporates at the C-terminus of nascent polypeptides. Thus, treatment with puromycin results in premature termination of translation. The generated puromycylated polypeptides can easily be traced with an anti-puromycin antibody. Immunoprecipitation confirmed that a portion of these nascent puromycin-containing polypeptides was ubiquitinated and removed in both WT and in Hi20S cells (Fig. 6g). Interestingly, the overall clearance of the puromycin-conjugates was faster in Hi20S than in control cells. However, the polyubiquitinated portion of these conjugates was removed faster in control cells. We conclude that a portion of inherently disordered proteins is removed by 20S proteasomes regardless of their ubiquitination status, although removal of unmodified proteins is faster.

A feature of 20S proteasome activity, as demonstrated in vitro, was the ability to proteolyze conjugated ubiquitin. If this occurs in vivo, an abundance of free 20S complexes in cells should

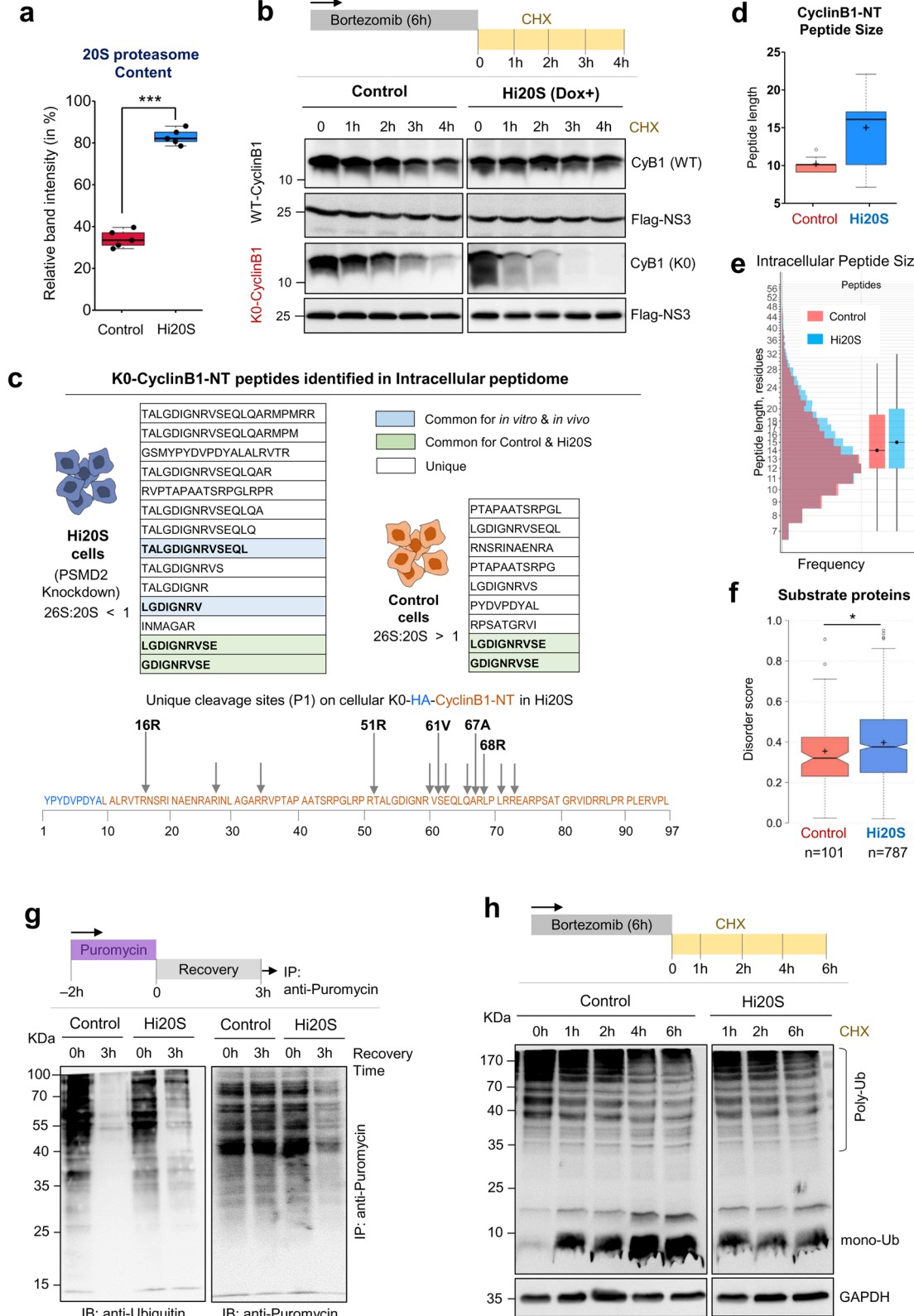

influence the extent to which ubiquitin molecules are degraded. To test this hypothesis, we monitored the depletion of polyubiquitin conjugates in WT and Hi20S cells. Although the steady-state ubiquitin landscape was higher in Hi20S than in control cells (Supplementary Fig. 14a, b), upon release from proteasome inhibition and arrest of protein synthesis, the total amount of polyubiquitin conjugates decreased in Hi20S cells with

no concomitant increase in free (unconjugated) ubiquitin (Fig. 6h). In contrast, the control cells accumulated unconjugated ubiquitin over time (Fig. 6h). Depletion of ubiquitin in Hi20S cells was not due to changes in gene expression as RT-PCR detected no significant changes in the transcription of UBA52 and UBC, and only a modest decrease in the transcription of UBB and RPS27A (Supplementary Fig. 14c). The 26S proteasome with its

**Fig. 6 The 20S proteasome proteolytic signature in cells. a** Quantification of 20S proteasome levels in control and Dox-induced Hi20S cells. Data represents the average 20S proteasome content calculated from anti-PSMA1 IB of five independent experiments (±SD error bar). $n = 5$ sample points. *** represents $p$ value < 0.0001 from unpaired $t$ test. **b** Control and Hi20S cells transiently expressing either HA-tagged WT- or K0- Cyclin B1-NT were pulsed with 0.5 µM bortezomib for 6 h followed by cycloheximide chase. Residual Cyclin B1-NT was detected by IB. **c** Tables list all peptides of K0-Cyclin B1-NT identified from intracellular peptidomics of Hi20S and control cells, respectively. Peptides highlighted in light-blue match to peptides obtained from in vitro cleavage of unmodified Cyclin B1-NT by isolated 20S proteasomes (as identified in Fig. 2). The light-green highlighted peptides were identified in both cell types. Unique cleavage sites identified on K0-Cyclin B1-NT in Hi20S cells are marked by arrows (illustration below). The corresponding P1 positions match the preferred P1 positions for 20S proteasomes cleavage of unmodified Cyclin B1-NT identified in vitro (as in Fig. 2). **d** Size distribution of intracellular Cyclin B1-NT derived peptides obtained from control and Hi20S cells (from three data sets each cell line). $n = 13$, 50 sample points. **e** Size distribution of all intracellular peptides obtained in control or Hi20S cells from triplicate data sets. Boxplot on the right represents median peptide size. Box limits indicate the 25th and 75th percentiles as determined by R software; whiskers extend 1.5 times the interquartile range from the 25th and 75th percentiles. **f** Disorder score distribution of potential proteasome substrates unique to each cellular condition (control, Hi20S). Proteins were assigned from intracellular peptide captured and scored based on intrinsic disordered elements. Control $n = 101$ and Hi20S $n = 787$ over 3 independent experiments. * represents $p$ value = 0.0215 from unpaired $t$ test. **g** Control and Hi20S cells were pulsed with puromycin for 2 h. Residual puromycin-containing polypeptides were immunoprecipitated at 0 and 3 h of recovery, and IB for ubiquitin and puromycin content. **h** Control and Hi20S cells were pulsed with 0.5 µM bortezomib for 6 h and chased with cycloheximide for up to 6 h of recovery; samples taken for ubiquitin IB. For box plots **a**, **d**, and **f**, the center lines (with/without notch) show the medians; box limits indicate the 25th and 75th percentiles as determined by R software; whiskers extend 1.5 times the interquartile range from the 25th and 75th percentiles, outliers are represented by open circles; crosses represent sample means. Source data are provided as a Source data file.

---

associated DUBs can release conjugated ubiquitin prior to proteolysis of the target protein[4], whereas the ability of the 20S proteasome to degrade the attached ubiquitin could explain the absence of free ubiquitin buildup in the Hi20S cell line.

**Signature activities of 20S proteasomes are apparent under hypoxia and in a failing heart.** As our results suggested that a modified ratio of 20S-to-26S proteasomes in cells should influence both the cellular proteome and the ubiquitin landscape, we explored the physiological conditions that could shift the balance between 26S and 20S proteasomes. When documenting the proteasome content in cultured cell lines we found that HeLa cells growing under hypoxia (1% O$_2$) showed a time-dependent shift from a majority of 30S/26S proteasomes to primarily 20S species (Fig. 7a, b). Since the total content of proteasome subunits in these cells did not change in response to hypoxia (Fig. 7c), this phenomenon was likely due to the disassembly of 30S/26S proteasomes releasing 20S core particles (Fig. 7d, e). Thus, we investigated whether these 20S complexes act as proteases. The turnover rates of both K0-Cyclin B1-NT and WT-Cyclin B1-NT were markedly different under hypoxia when compared to normoxic cells. Whereas proteolysis of non-ubiquitinated K0-Cyclin B1-NT was faster under hypoxia, WT-Cyclin B1-NT was degraded slower in hypoxic than in normoxic cells (Fig. 7f, g). This result is completely aligned with the potential role of 20S proteasomes in degrading non-ubiquitinated cyclin B1 demonstrated above. We ruled out the possible involvement of alternative proteasome activators[55,56] since no significant association of PA28 or PA200 with free 20S complexes was detected under the experimental conditions (Supplementary Fig. 15a–c). Accelerated degradation of non-ubiquitinated misfolded proteins alongside the elevated levels of 20S proteasomes documented in hypoxic cells substantiates the potential role of the 20S proteasome as an active cellular protease in cells.

The higher ratio of 20S-to-26S proteasomes observed in both HeLa and HEK293 cells under hypoxia (Figs. 8a and 7e) suggests that detecting 20S signature proteolytic activities in hypoxic cells is plausible. Capturing the total cellular peptidome, sequencing the peptides using MS/MS (summarized in Supplementary Data 9), and determining the peptide sizes in hypoxic cells demonstrated longer peptide sizes in hypoxic vs normoxic cells (Fig. 8b). Moreover, the peptides from the hypoxic cells matched proteins with a significantly higher median disordered score relative to peptides from normoxic cells (Fig. 8c).

A key signature of the 20S proteasome identified in vitro was their capacity to proteolyze conjugated ubiquitin along with a misfolded target. As hypoxic cells displayed a higher ratio of 20S-to-26S proteasomes, we investigated the 20S proteasome-associated ubiquitin degradation during hypoxia. First, we documented the ubiquitination pattern in cell lysates during hypoxia. Polyubiquitin conjugates increased for the first 12 h of hypoxia, but then decreased over time up to 36 h without concomitant generation of free Ub (Fig. 8d, left). Notably, it was lysine48-linked polyubiquitinated conjugates that were removed, rather than lysine63-linkages (Fig. 8d, right). We confirmed that this decrease in polyubiquitinated conjugates was due to degradation and not a loss-of-function of E1 activity (Supplementary Fig. 15d). Depletion of the cellular ubiquitin pool during adaptation to hypoxia was concomitant with an increased abundance of 20S proteasomes.

A pathological condition of prolonged hypoxia is an ischemia-associated end-stage failing heart, known to activate several proteolytic pathways[57,58]. We tested myocardial tissue samples of human individuals with failing heart and found an abnormally high ratio of 20S-to-26S proteasomes (Fig. 8e, Supplementary Fig. 16a) similar to the ratio observed in cultured HeLa and HEK293T cells under hypoxia (Figs. 8a and 7e). These failing heart tissues also displayed elevated polyUb-conjugates levels relative to normal heart tissue (Supplementary Fig. 16b). We then performed whole tissue peptidomics using the same approach as described above and obtained evidence for the proteolysis of ubiquitin in failing heart tissues (Fig. 8f). Five ubiquitin peptides were identified in myocardial tissue (Supplementary Fig. 16c) with a significantly higher integrated intensity in failing than in normal heart tissue (Supplementary Fig. 16d).

To validate whether the proteasome is the source of the ubiquitin-derived peptides found in heart tissues, we developed an approach to identify proteasome trapped peptides (PTPs) by rapidly isolating intact proteasome complexes directly from the tissue and extracting peptides for MS/MS sequencing (Supplementary Fig. 17a, b, Supplementary Data 10, 11). The integrated intensity of ubiquitin-derived peptides associated with these proteasomes of the failing heart was over two orders of magnitude greater than ubiquitin PTPs from a normal heart (Supplementary Fig. 17c, d). Notably, these ubiquitin-derived peptides identified among PTPs were the same as those identified in the total intracellular peptidome (Supplementary Fig. 16c, d). Furthermore, whole tissue peptidomics detected a larger repertoire of potential 20S proteasomes substrates in failing heart samples (Supplementary Fig. 18a, b). Activated caspases and calpain

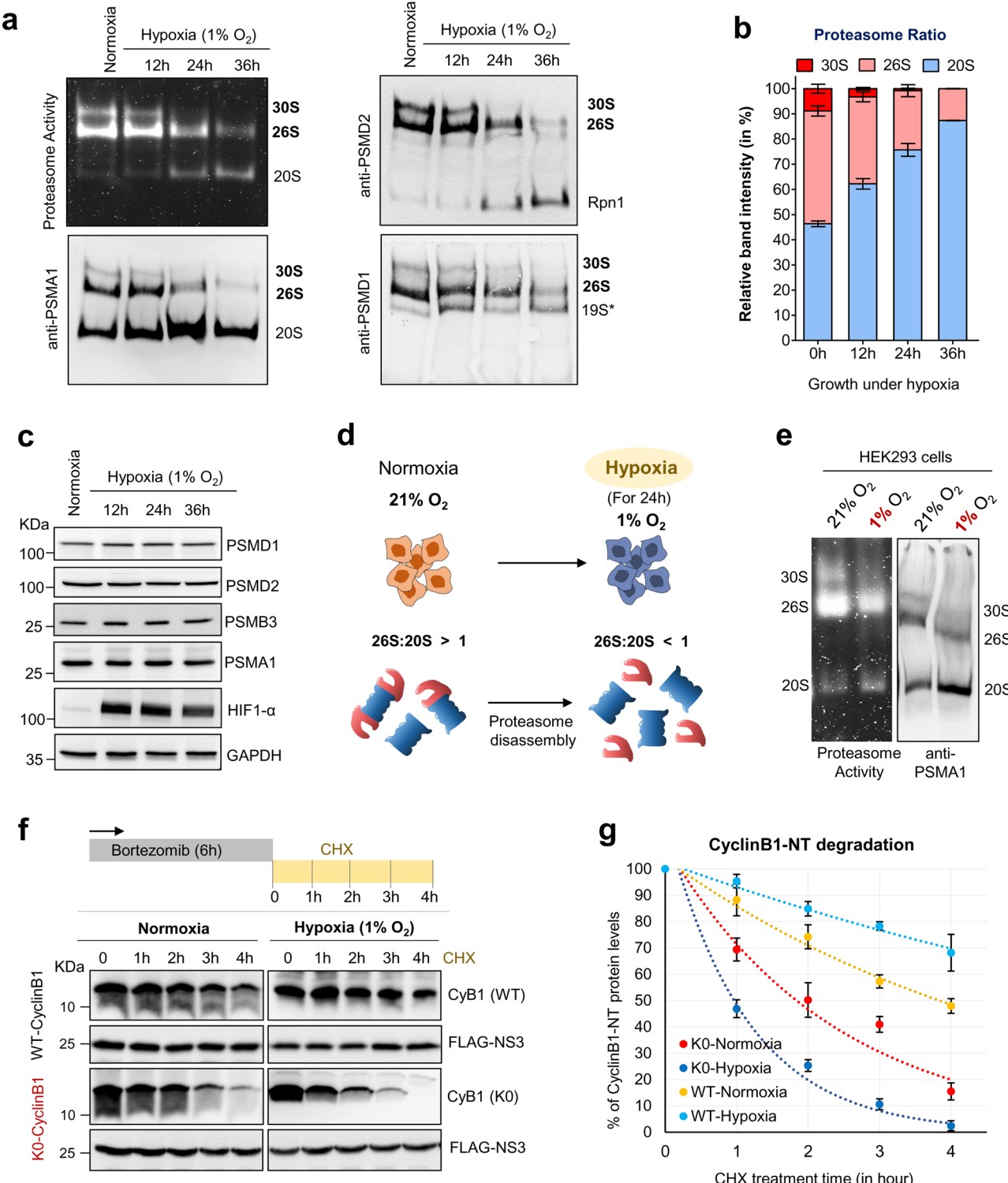

under ischemic conditions[57] may also contribute to intracellular proteolysis, nonetheless, elevated 20S proteasomes and the higher disordered score of substrate proteins reflected in the total intracellular peptidome of the failing heart (Fig. 8g) is in agreement with the signature properties of 20S proteasomes.

**The 20S proteasome contributes to cell survival**. We hypothesized that the presence of 20S proteasomes, which can proteolyze disordered proteins faster than the 26S proteasome, could confer survival benefits to cells, when damaged/misfolded proteins need to be removed. Therefore, we evaluated the contribution of free 20S proteasomes to cell survival under stress associated with damaged/misfolded proteins, by comparing the endurance of Hi20S and control cells to conditions that induce protein misfolding (puromycin exposure). Hi20S cells showed better survival than control cells at all puromycin concentrations (Fig. 9a, b, Supplementary Fig. 19a). Cell survival correlated with faster removal of puromycylated polypeptides by proteasomes in Hi20S cells (Fig. 9c and Supplementary Fig. 19b, c).

**Fig. 7 Hypoxia-induced 20S proteasome levels and accelerated Cyclin B1-NT degradation. a** HeLa cells were grown under normoxia (21% O$_2$) or hypoxia (1% O$_2$) for the indicated time periods. Cell lysates were resolved by 4% native gel followed by proteasome in-gel activity assay or transferred for IB of native complexes using the indicated proteasome antibodies (the 19S RP subunit PSMD1, PSMD2 and 20S CP subunits PSMA1). **b** The calculated relative distribution of 30S, 26S, and 20S proteasome species in normoxic or hypoxic cells. Data represent the mean percentage of proteasome content (±SD error bar) calculated from PSMA1-native IB of three experiments at each time period. **c** The lysates of cells under hypoxia (as in **a**) were resolved by SDS-PAGE followed by IB to evaluate proteasome subunits (HIF1-α is a marker for hypoxia, GAPDH is a loading control). **d** The cartoon illustrates proteasome disassembly under hypoxia into 20S and 19S sub-complexes. **e** HEK293T cells were either grown under normoxia or hypoxia (1% O$_2$) for 24 h. Cell lysates were resolved by 4% native gel for proteasome in-gel activity assay or transferred for immunoblot (IB) using anti-PSMA1 antibody. **f** HEK293T cells grown under normoxia or hypoxia for 24 h were transfected with WT- and K0-Cyclin B1-NT plasmid constructs, then treated with 0.5 μM bortezomib followed by cycloheximide chase. Cyclin B1-NT levels were detected by IB. **g** Line graph represents WT- and K0-Cyclin B1-NT protein levels under conditions as in (**f**) measured from anti-cyclin B1 antibody IB, under both normoxia and hypoxia at each time point of cycloheximide treatment. Error bars (±SD) were calculated from the value of three independent experiments. Source data are provided as a Source data file.

We then evaluated whether the documented elevated levels of the 20S proteasomes under hypoxia contribute to cell survival under such conditions. To this end, we challenged hypoxic cells with puromycin and followed the clearance of puromycylated polypeptides. Interestingly, these polypeptides were cleared from hypoxic cells faster than from cells growing under normoxia (Fig. 9d, e). Although the rate of protein synthesis was somewhat attenuated following hypoxia (Supplementary Fig. 20a,b), the rate of removal was unaffected by differential puromycin labeling and did not account for the differences (Supplementary Fig. 20c,d). The removal of puromycylated polypeptides under hypoxia was attributed to proteasome function despite the diminished 26S proteasome content (Supplementary Fig. 20e) suggesting compensation by the elevated 20S complexes in these cells. Moreover, these hypoxic cells survived better at all the tested concentrations of puromycin compared to the same cells grown under normoxia (Fig. 9f and Supplementary Fig. 21a). Notably, in the absence of puromycin, this cell line grew at the same rate in hypoxic and normoxic conditions (Supplementary Fig. 21b). The resistance to low concentrations of puromycin when grown under hypoxia is compatible with the improved ability of Hi20S cells to survive in the presence of puromycin (Fig. 9b), ruling out hypoxia-related metabolic shift as the underlying cause. All these results support an active proteolytic role for the 20S proteasome and contribution to cell survival under hypoxia.

## Discussion

Eukaryotic cells typically contain a mixture of 26S (generally referring to both 26S and 30S species) and 20S proteasomes, though their ratios vary in different cell types or in response to environmental conditions. The 26S proteasome has been studied extensively and its role in ubiquitin-dependent proteolysis is well established. However, the role of the 20S core particle as a stand-alone protease in cells has been challenging to discern. Characterizing the properties of the 20S proteasome in vitro and then investigating them in cells provided evidence supporting the cellular role of 20S proteasomes as a functional proteolytic enzyme with distinct properties from those of the 26S proteasome, both in substrate selection and proteolytic outcome.

Many proteases contribute to the steady state of the peptidome by continuously generating and further hydrolyzing peptides. Therefore, it is challenging to identify the peptides in the cell that are produced directly by the proteasome. Initially we defined the distinct characteristic cleavage patterns of the 20S and 26S proteasomes by comparing their effects on identical sets of chemically synthesized substrates that contained both an unstructured region and lysine48-linked ubiquitin chains. Despite having identical catalytic sites, the proteolytic outcome of the two complexes differed, possibly due to the mechanism by which the substrate is recognized and translocated into the catalytic chamber[10,13]. The 20S proteasomes generated longer peptides

from a given substrate and a greater variety relative to the 26S proteasomes. Later this signature characteristic was also detected in the intracellular peptidome of cells with an elevated 20S-to-26S proteasome ratio. Earlier in vitro studies that measured proteolysis of disordered proteins also reported that isolated 20S complexes generated products with a longer mean size compared to 26S complexes (4-35aa vs 4-30aa[33] or 7-20aa vs 7-15aa[32]), however, the influence of ubiquitination on the product outcome has not been studied until now. In the current study, both proteasomes produced fewer and slightly shorter peptides as the polyubiquitin chain on the substrate increased in length. Longer peptides retain some secondary structure, that increases their potential to serve in signaling roles[59]. The inherent ability of the 20S proteasome to generate a greater variety of peptide products and potentially longer ones suggests that 20S activity may alter the intracellular peptidome in a manner that is beneficial to downstream signaling pathways such as the presentation of antigenic peptides to the immune system[60].

One of the unexpected outcomes of the 20S proteasome activity was the degradation of the ubiquitin tag along with an unstructured conjugate. It is plausible that ubiquitin can be unfolded, as both Cdc48[61] and 26S proteasomes[4] unfold ubiquitin conjugated to substrates. Direct and functional coupling between the archaeal or yeast Cdc48 and the 20S has been documented in vitro[24,39]. Therefore, it is possible that in cells, ATPase complexes such as Cdc48/p97 or other chaperones partially unfold the substrate thereby preparing it for degradation by the 26S or even by the 20S proteasome. However, in the current study, we found that isolated 20S complexes lacking ATPase activity proteolyzed a tightly folded globular domain such as ubiquitin when attached to a largely disordered protein, without any auxiliary chaperone or proteasomal activator. A possible explanation may be that the unstructured polypeptide serves as an initiation site for recognition by the 20S proteasome[13]. Although free 20S complexes are often found in a closed-gate latent state, some proteins with substantial disorderedness are effective in activating their own degradation by gating the 20S[42]. Upon engagement, pulling of the polypeptide through the narrow aperture of the 20S α-ring leads to one-by-one breaking of hydrogen bonds, bypassing the initial force barrier required to overcome the unfolding of a globular protein tertiary structure (Fig. 10). The force required to unfold a protein as it is pulled through a pore is much smaller than that of the simultaneous unfolding of a globular domain, as occurs in chaperonins[62]. Initial engagement of a ubiquitin conjugate would be by the disordered polypeptide (e.g., cyclin B1) that induces flexibility in the central pore into the 20S α-ring. If so, the disordered initiation site must be long enough to reach the β-protease active sites[63] that provide the pulling force through the α-ring aperture. Another explanation may be that substrate engagement induces sufficient flexibility in the pore region for the antechamber to engulf the conjugated ubiquitin where surface

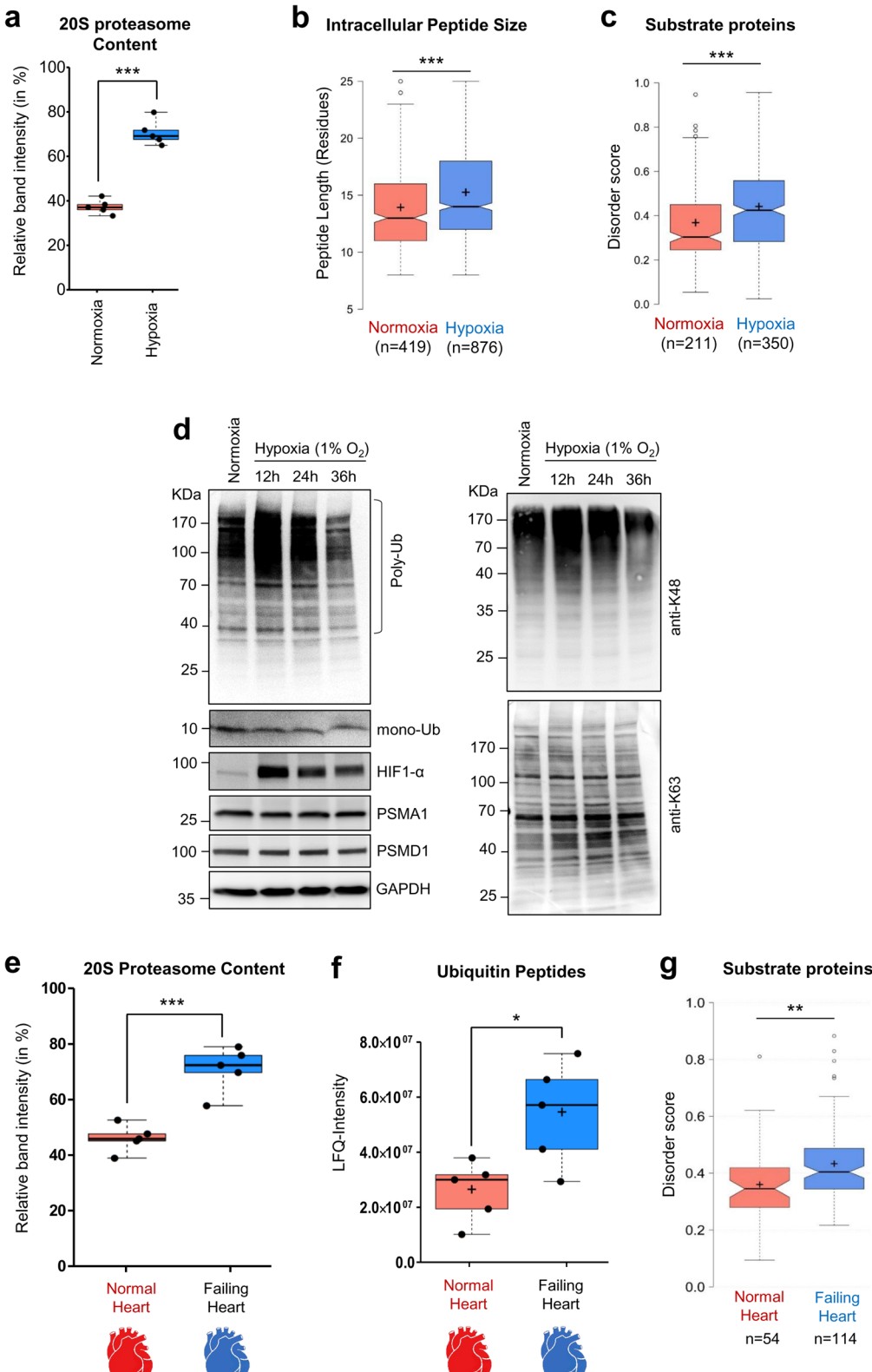

contacts provide local secondary structure melting. Indeed, we observed binding of a ubiquitinated substrate induced discontinuous density and conformational dynamics in the center of the 20S α-ring of sufficient dimensions to potentially enable simultaneous entry of a few polypeptides. Unfolding may even occur in two steps: partial unfolding as ubiquitin traverses the outer pore in the α-ring and enters the antechamber, followed by

secondary unfolding through the next aperture defined by the β-annulus bordering the catalytic chamber. The finding that 20S substrates are not limited to fully unfolded or completely unstructured proteins suggests that 20S proteasomes play a larger role in controlling the proteome than previously thought. Specifically, disordered segments may serve as 20S degrons shunting certain proteins (even those with limited globular domains) to the

**Fig. 8 Signatures of 20S proteasomes observed in hypoxia and human failing heart. a** The ratio of 20S-to-26S proteasomes in HeLa cells grown under normoxia or hypoxia. Data represent the average proteasome content calculated from IB using anti-PSMA1 of five independent experiments (±SD error bar). *** represents *p* value < 0.0001 from unpaired *t* test. **b** Average size distribution of intracellular peptides obtained from HeLa cells grown under normoxia and hypoxia (*n* = 419, 876 sample points over 3 independent experiments). *** represents *p* value < 0.0001 from unpaired *t* test. **c** Protein disorder score distribution of potential proteasome substrates unique to each cellular condition (normoxia or hypoxia). Proteins were assigned from captured intracellular peptides, and scored based on intrinsic disordered elements; normoxia *n* = 211 and hypoxia *n* = 350 over 3 independent experiments. *** represents *p* value < 0.0001 from unpaired *t* test. **d** Polyubiquitin content of HeLa cells grown under normoxia or under hypoxia. IB using antiUb, anti-K48, and anti-K63 antibodies. **e** Proteasome content of human heart-muscle tissue under failing (*n* = 5) or normal (*n* = 5) conditions. The boxplot represents average % of the 20S proteasome content measured from anti-PSMA1 probed native IB; *n* = 5 sample points. *** represents *p* value = 0.0004 from unpaired *t* test. **f** Ubiquitin-derived peptides in heart-muscle tissues. Human frozen heart-muscle tissue (failing, *n* = 5; normal, *n* = 5) was subjected to intracellular peptide isolation and analyzed quantitatively for ubiquitin peptides by LC-MS/MS. The plot represents the LFQ-intensity of ubiquitin calculated from its peptides in each sample (*n* = 5). * represents *p* value = 0.0204 from unpaired *t* test. **g** Protein disorder score distribution of potential proteasome substrates unique to each heart sample set. Proteins were assigned from intracellular peptide captured and scored based on intrinsic disordered elements; non-failing heart *n* = 54 and failing heart *n* = 114 over 5 heart tissue samples each. ** represents *p* value = 0.0010 from unpaired *t* test. For box plots **a**, **b**, **c**, **e**, **f**, and **g**, the center lines (with/without notch) show the medians; box limits indicate the 25th and 75th percentiles as determined by R software; whiskers extend 1.5 times the interquartile range from the 25th and 75th percentiles, outliers are represented by open circles; crosses represent sample means. Source data are provided as a Source data file.

20S proteasome for removal. Whereas such proteins could potentially be substrates of both 20S and 26S proteasomes, the less regulated and simpler 20S complexes may provide added value under specific conditions.

The ubiquitous and rather abundant 20S core particle alongside the 26S proteasome holoenzymes in most cell types (in the current study estimated at around 40%) has long baffled researchers. With the understanding that the 20S proteasome is a functional enzyme, its presence in cells does not represent merely unassembled proteasome intermediates or released core particles from disassembled holoenzymes, but rather provides a gain-of-function. As demonstrated in the current study, elevated 20S proteasome levels correlate with efficient removal of damaged proteins, providing benefits during stress (e.g., hypoxia or stress imposed by synthesis of truncated nascent polypeptides). Long-term stress such as prolonged hypoxia, or in human failing heart tissue, may result in a higher 20S-to-26S proteasome ratio to unleash the signature activities of free 20S enzymes. While the degradation of ubiquitin conjugates is essential for cell viability, the necessity to rapidly relieve the proteotoxicity caused by defective/damaged polypeptides may come at the expense of the small portion of conjugated ubiquitin that is degraded along with the target substrates by 20S proteasome activity. Elevated levels of 20S complexes, often thought to reflect a breakdown of proteasome holoenzymes, is a characteristic of aging cells[64–66]. However, these elevated levels may be more than a marker for aging, as 20S proteasomes also contribute to survival during the aging process, which is characterized by diminished protein quality control systems.

## Methods

**Plasmids and cloning**. The DNA fragment of the N-terminal part of cyclin B1 (1–88 aa) with HA-tag (HA-Cyclin B1-NT) was cloned into pCMV10-3XFlag vector. Both wild type and all lysine to arginine (K0) mutant fragments were used for cloning (Fig. 1a). To express NS3 fused HA-Cyclin B1-NT in mammalian cells, we fused NS3Pro at the N-terminus of HA-Cyclin B1-NT by designing two DNA fragments (Supplementary Methods) and cloning into EcoRI and HindIII sites of pYFP-SmaSh vector (Addgene: 111500). pTRIPZ-PSMD2 shRNA plasmid for human cell expression was a kind gift from Peter Tsvetkov, Broad Institute.

**Chemical synthesis and ubiquitination of conjugates**. The detailed step-by-step synthesis of the ubiquitin conjugates is available in Supplementary Methods and Supplementary Figs. 22–33. General Methodology: SPPS was carried out manually in syringes, equipped with teflon filters, purchased from Torviq, or by using an automated peptide synthesizer (CSBIO). Analytical HPLC was performed on an instrument (Thermo Scientific, Spectra System p4000) using an analytical column (Xbridge BEH300 C4 3.5 μm, 4.6 × 150 mm) at a flow rate of 1.2 mL/min. Pre-parative HPLC (Waters) was performed using X Select C18 10 μm 19 × 250 mm and semi-preparative HPLC (Thermo Scientific, Spectra System scm1000) was

performed using Jupiter C4 10 μm, 300 Å, 250 × 10 mm column, at flow rates of 15 and 4 mL/min, respectively. Commercial reagents were used without further purification: Analytical grade DMF (Biotech), Resins (Creosalus), protected amino acids (GL Biochem), and activating reagents HBTU, HOBt, HCTU, HATU (Luxembourg Bio Technologies). Buffer A: 0.1% TFA in water; Buffer B: 0.1% TFA in acetonitrile.

**Purification of human proteasomes**. To obtain human proteasomes[5], human erythrocytes (RBCs purchased from The Israel blood bank) were washed with chilled 1X PBS and lysed in hypotonic buffer (25 mM Tris pH 7.4). The cell debris was separated by centrifugation and the clear red cell lysate was loaded onto 50 mL DEAE Affigel blue column (Bio-Rad) and washed with Buffer A (25 mM Tris pH 7.4, 10% glycerol, 10 mM MgCl2, 1 mM ATP, 1 mM DTT). Proteasomes were eluted with a gradient elution from 0 to 50% of Buffer B (25 mM Tris pH 7.4, 10% glycerol, 10 mM MgCl2, 1 mM ATP, 1 mM DTT with 1 M NaCl). The active fractions were then checked by 25 μM Suc-LLVY-AMC activity assay[67,68], pooled, and loaded onto 8 mL Resource-Q column with Buffer A. Then, proteasomes were eluted with a gradient elution from 0 to 50% of Buffer B and active fractions were examined by activity assays conducted either in a 96-well plate or native PAGE. To obtain the highest purity, gel filtration was performed for each fraction using a S300 (120 mL) column (GE Life Sciences) with Buffer A. The purity and integrity of the proteasomes were evaluated in native PAGE and SDS-PAGE, then concentrated, aliquoted, flash-frozen in liquid nitrogen, and stored at −80 °C.

**Enzymatic ubiquitination and separation of K48-Ub chains**. Recombinant Monomeric Ub, Ub$^{S20C}$, E2 conjugating enzymes, and human E1 were purified[69]. For fluorescent-tagged Lysine48-Ub chain preparation, Ub$^{S20C}$ was labeled with Alexa Flour 488 attached to the Cysteine20 residue[70]. To reduce cysteine thiol groups, 60 nmol of the ubiquitin variant was incubated with 120 nmol TCEP in 400 mL PBS (pH 7.5) for 10 min at 25 °C. The labeling preceded 90 min at 25 °C in the dark with a fourfold excess of fluorescent dye. 10 mM β-mercaptoethanol (BME) was added to stop the reaction. Desalting steps (Nap5 columns, GE-Healthcare) into 20 mM HEPES/NaOH (pH 7.5) removed excess dye and BME. The eluate was concentrated (Amicon Ultra-0.5, 3 kDa cutoff, Millipore), and aliquots were stored at −80 °C. Enzymatically synthesized Lysine48-Ub chains were obtained from a reaction containing 1 mM of Ub or Ub-His6, 80 nM E1 (UBA1), 40 mM E2-25K, 4 mM TCEP, and 15 mM ATP in a volume of 1 mL with 50 mM Tris pH 8.0 buffer incubated at 37 °C for 20 h[71]. Then, the reaction mixture was resolved by SEC-70 10/300 (Bio-Rad) in Tris pH 7.4 buffer with 150 mM NaCl and 1 mM DTT. The ubiquitin chain fractions were confirmed by SDS-PAGE and fractions with shorter chains (≤4 Ub) were pooled and acidified with a final 50 mM NH4 Acetate pH 4.0. Subsequently, the mixture was loaded into Resource-S (6 mL) column with cationic exchange buffer A (50 mM NH4 Acetate 4.5 pH), eluted with a gradient step from 10 to 30% buffer B (50 mM NH4 Acetate 4.5 pH with 1 M NaCl) for 20 column volumes and the purity of the chains was confirmed by SDS-PAGE and proteins were concentrated, aliquoted, and stored at −20 °C. To obtain the highest purity, gel filtration was performed for each fraction using a SEC-70 10/300 (Bio-Rad) in Tris pH 7.4 buffer with 150 mM NaCl and 1 mM DTT.

**Cell culture and growth conditions**. HEK293T (ATCC), and HeLa cells (ATCC) were grown in HiGlucose DMEM media (Sigma) supplemented with 10% FBS (Biological Industries) and antibiotics in 5% CO2 incubator at 37 °C. For hypoxia treatment, cells were grown in the above media in a hypoxia chamber (Baker Ruskinn) with 1% O2 and 5% CO2 at 37 °C. All transient transfection was performed using the X-tremeGENE HP transfection reagent (Roche). T-47D PSMD2-

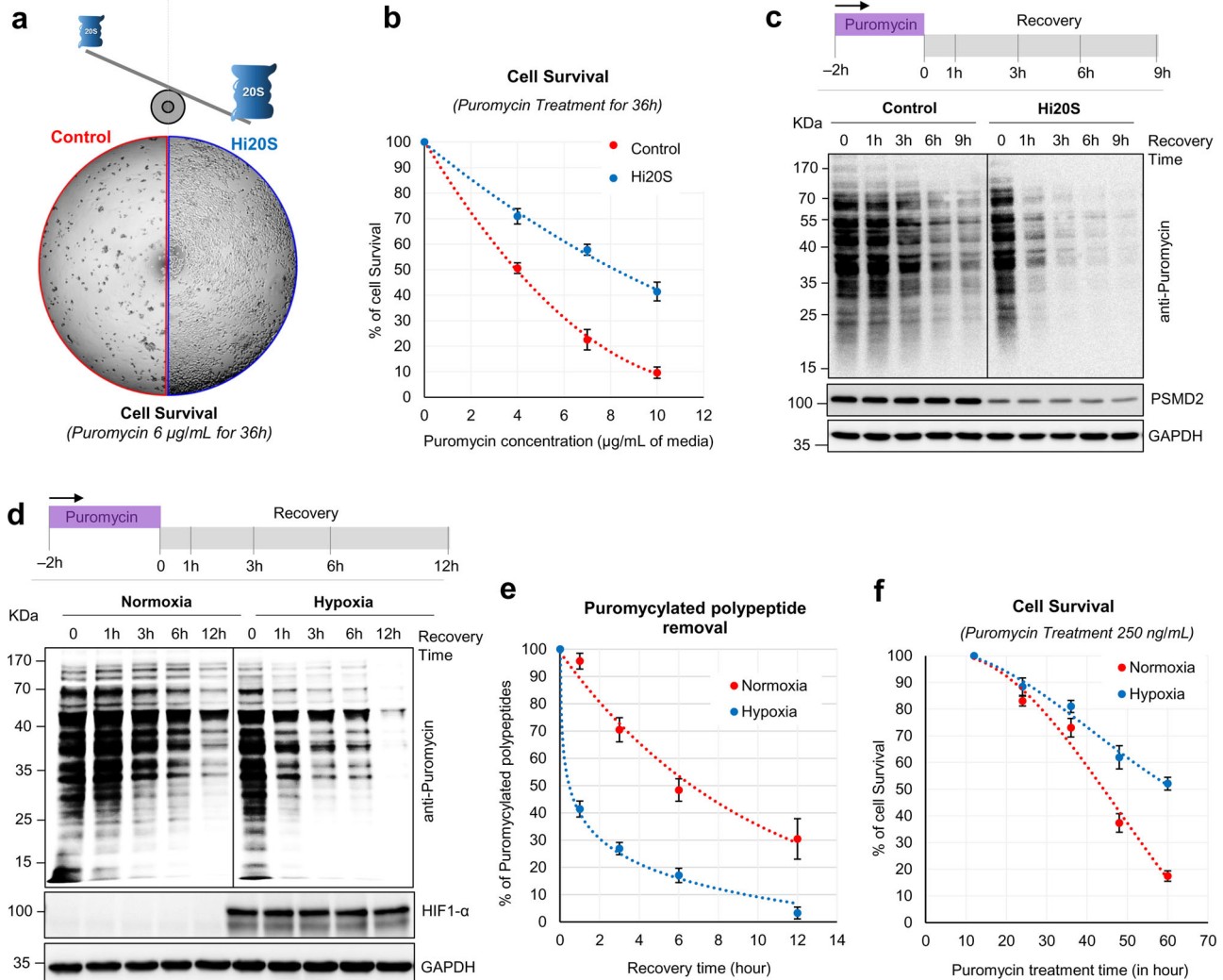

**Fig. 9 Elevated 20S proteasome levels (Hi20S cells) facilitate survival under proteotoxic stress. a** Phase contrast image of culture plate showing survival of control and Hi20S cells after 36 h of puromycin treatment (6 µg/mL). **b** Cell survival at increasing puromycin exposure. Control and Hi20S cells were grown for 36 h in the presence of increasing puromycin concentrations. Cell survival was quantified by the MTT assay. Data represent the average of three experimental values (±SD error bar). **c** Control and Hi20S cells were treated with puromycin (5 µg/mL) for 2 h and then recovered over 12 h. Cell lysates were subjected to IB with anti-puromycin antibody. **d** HeLa cells were grown under hypoxia and treated with puromycin as described in (**c**) IB was performed with anti-puromycin antibody. **e** The rate of puromycylated polypeptides removal under normoxia and hypoxia. Data represent the average percentage of puromycin-containing polypeptides (±SD error bar) at each time point of the recovery phase, quantified from IB of three independent experiments. **f** Cell survival upon puromycin exposure during hypoxia. HeLa cells were grown under normoxia or hypoxia with puromycin (250 ng/mL) for up to 60 h. Cell survival was quantified by the MTT assay. Data represent the average of three experimental values (±SD error bar). Source data are provided as a Source data file.

knockdown cell line (a kind gift from Peter Tsvetkov, Broad Institute) was grown in RPMI media (Sigma) supplemented with 10% FBS (Biological Industries) and antibiotics in 5% $CO_2$ incubator at 37 °C. To generate doxycycline-inducible PSMD2 knockdown clones in HEK293T cells, linearized pTRIPZ-shRNA_PSMD2 (From 5′LTR to 3′LTR) was transfected followed by clone selection under puromycin at 1 µg/mL concentration. High RFP expressing, puromycin resistant cells were selected by FACS and grown at 0.5 µg/mL puromycin to generate stable clones. For induction of PSMD2 shRNA expression, the stable clone cells were grown with doxycycline at 1 µg/mL concentration for three passages. For various experiments, different reagents, e.g., bortezomib (0.5 or 1 µM), chloroquine (50 µM), E1 inhibitor PYR-41 (10 µM), cyclohexamide (50 µg/mL), the amino-peptidase inhibitor CHR-2797 (1 µM), E64d (10 µM), Z-VZD-FMK (50 µM), and puromycin (250 ng/mL to 10 µg/mL) were used.

**Lysis of mammalian cell culture and Immunoblotting**. Cell culture plates were washed twice with warmed 1x PBS and cell lysates were prepared by adding chilled NP-40 Lysis buffer (50 mM Tris 7.4 pH, 150 mM NaCl, 1% NP-40, 1x protease inhibitor cocktail, 5 mM iodoacetamide) to the culture plates. The lysate content was collected in small centrifuge tubes and incubated on ice for 30 min followed by centrifugation at 4 °C/13,000 × g. The supernatant lysate was collected and stored at

−20 °C until further use. Denaturing/Native immunoblotting was carried out using various antibodies: anti-HA (Sigma-SAB1305536-40TST) in 1:2000, anti-HA-HRP (R&D Systems-HAM0601) in 1:1000, anti-Myc (Biolegend-626802, Clone-9E10) in 1:2000, anti-FLAG (Biolegend-637301, Clone-L5) in 1:2000, anti-PSMA1 in 1:5000, anti-Cyclin B1 N-terminal (abcam-ab226397) 1:5000, anti-αTubulin (Biolegend-627902, Clone 10D8) in 1:2000, anti-MCL-1 (abcam-ab32087) in 1:1000, anti-GAPDH (Sigma-G9545) 1:5000, monoclonal anti-ubiquitin (Covance-MMS-257P-200, Clone P4D1) in 1:5000, polyclonal anti-ubiquitin (Dako-Z0458) in 1:3000, anti-LC3B (abcam-ab51520) in 1:1000, anti-puromycin (Millipore-MABE343, Clone 12D10) in 1:5000, anti-PSMD2 (Bethyl-A303-853A) in 1:1000, anti-PSMD1 in 1:3000, anti-PSMB3 in 1:1000, anti-HIF1-α (abcam-ab179483) in 1:1000, anti-K48 (Millipore-05-1307, Clone Apu2) in 1:1000, anti-K63 (Millipore-05-1308, Clone Apu3) in 1:1000, anti-PSMD4 in 1:1000, anti-PSMC4 (proteintech-11389-1-AP) in 1:1000, anti-PSMC3 in 1:1000, anti-USP14 (Sigma-WH0009097M4) in 1:1000, anti-PA200 (abcam181203) in 1:1000 and anti-PA28α (Cell Signaling-2408) 1:1000.

**Immunoprecipitation for puromycin-containing polypeptides and ubiquitin conjugates**. The cell lysates were prepared as above with a minor change in NP-40 concentration (0.4%). Then pre-cleared cell lysates were prepared by incubating

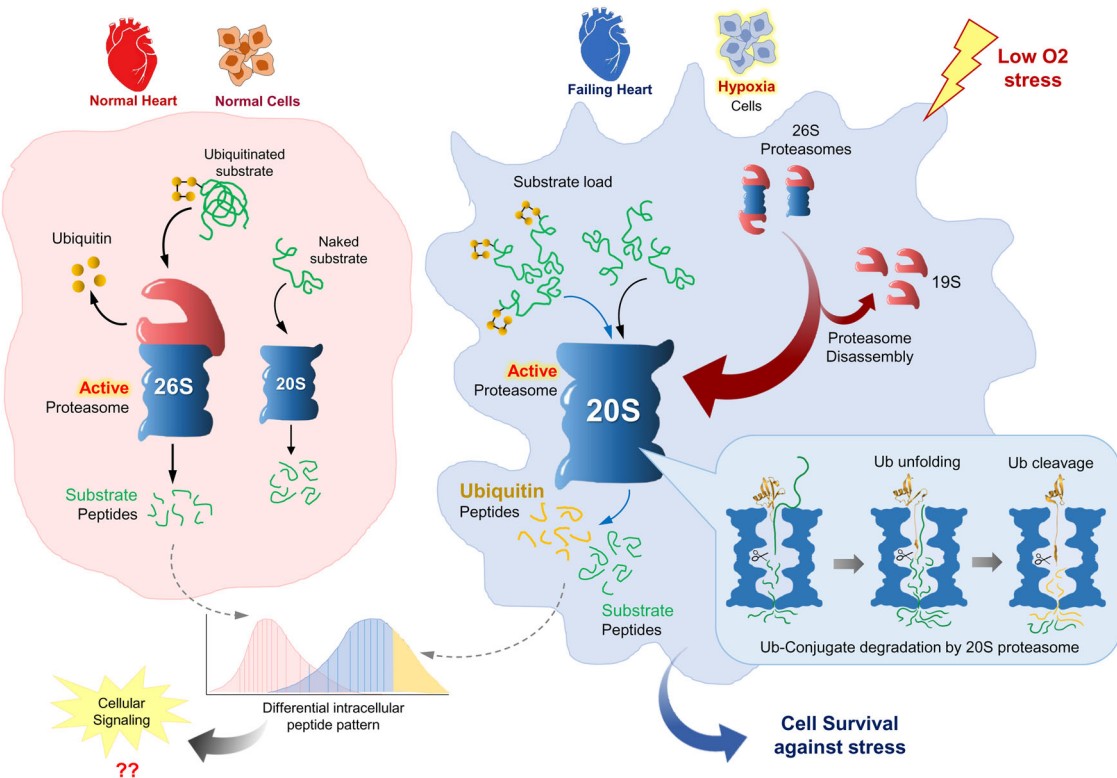

**Fig. 10 A model demonstrating the putative contribution of 20S proteasomes to proteolysis during hypoxia.** Under normoxia, the 26S proteasome degrades proteins and recycles the conjugated ubiquitin tag. Disassembly of 26S proteasomes under hypoxia leads to elevated levels of free 20S core particles (CP). Excess-free 20S CP are proposed to serve as active proteasomes capable of degrading a portion of ubiquitin along with the conjugated substrate. The disordered segment of the substrate inserts into the gate at the center of the 20S α-ring and accesses the proteolytic active sites, until the ubiquitin domain is localized to the pore. Stepwise protease-driven unfolding of ubiquitin brings the conjugation region to the vicinity of the β-proteolytic active sites. Ubiquitin is proteolyzed and peptide products from both substrate and ubiquitin are released, occasionally including a remnant of ubiquitin still linked via an isopeptide bond to a substrate-derived peptide.

with protein A/G conjugated agarose beads for 30 min at 4 °C with slow rotation. 1 mg pre-cleared cell lysates were then added to 20 μL of fresh Protein A/G conjugated agarose beads along with 4 μg of antibodies (anti-Puromycin/anti-Ubiquitin). The mixture was incubated for 2 h at 4 °C with slow rotation followed by five 1 mL washes with chilled wash buffer (50 mM Tris 7.4 pH, 300 mM NaCl, 1x protease inhibitor cocktail, 5 mM iodoacetamide). The target conjugates were then eluted from beads by mixing protein loading dye (without reducing agents BME/DTT) and heating for 2 min at 95 °C followed by denaturing immunoblot.

**Native cell lysis for proteasome activity.** Cultured cells were collected from plates by trypsinization and lysed using ATP buffer (25 mM Tris 7.4 pH, 10% glycerol, 10 mM MgCl$_2$, 1 mM ATP, 1 mM DTT) with glass beads and vortexing at 4 °C. Following centrifugation, the clear supernatant (native lysate) was collected, aliquoted, flash-frozen in liquid nitrogen, and stored at −80 °C.

**Native gel, in-gel activity assay, and native immunoblotting.** Native lysates were run in 4% native PAGE in native running buffer (100 mM Tris Base, 100 mM Boric acid, 1 mM EDTA, 2.5 mM MgCl$_2$, 0.5 mM ATP, 0.5 mM DTT) for 2 h at 130 V. For In-gel activity assay, the gel was incubated in ATP buffer with 25 μM Suc-LLVY-AMC at 37 °C for 10 min and imaged under UV light. For Native immunoblotting, the gel was soaked in SDS-running buffer for 20 min and then transferred onto PVDF membrane.

**In vitro substrate degradation assay.** The degradation assay included 20 nM of human 26S or 20S proteasomes and unmodified Cyclin B1-NT or ubiquitinated Cyclin B1-NT conjugates in a molar ratio of 1:150. The 26S proteasome assay buffer contained 25 mM TRIS (pH 7.4), 10 mM MgCl$_2$, 10% glycerol, 1 mM ATP, and 1 mM DTT. The 20S proteasome assay buffer contained 25 mM TRIS (pH 7.4), 150 mM NaCl, 10% glycerol without MgCl$_2$ and ATP. All degradation reactions were carried out at 37 °C and were terminated by adding SDS-loading dye.

**Peptide isolation from in vitro reactions.** For isolating peptides from in vitro reactions, the aforementioned proteasome-substrate reactions were terminated by heating mixtures at 95 °C for 5 min. Then, the mixture was passed through 30 kDa

cutoff Amicon® Ultra - 0.5 mL Centrifugal Filter unit (Merck-Millipore). The flow-through was collected and desalted by C18 StageTip column.

**Intracellular peptide isolation.** The cultured cells were collected from the plates by trypsinization, then lysed by adding 80 °C hot water and incubating the mixture at 80 °C for 20 min[72]. Subsequently, the lysate was centrifuged at high speed (16,000 × g) for 30 min and the supernatant was kept at −80 °C overnight. The lysate was then acidified with HCl to a final concentration of 0.01 N HCl and incubated on ice for 30 min. After 20 min centrifugation at 16,000 × g the pellet was discarded and the supernatant was passed through 30 kDa cutoff Amicon® Ultra - 0.5 mL Centrifugal Filter unit (Merck-Millipore). The flow-through was collected and desalted by C18 StageTip column.

**Human heart tissue procurement.** Ventricular myocardial tissue from patients with advanced heart failure (failing heart) at the University of Michigan was collected at the time of cardiac transplantation[73]. Non-failing ventricular myocardial tissue was collected from unmatched donors from the University of Michigan and other regional hospitals through a protocol with the Gift of Life Michigan Organ and Tissue Donation Program. This project was approved by the University of Michigan Institutional Review Board (IRB) and subjects gave informed consent. Each heart was perfused with ice-cold cardioplegia, samples were snap-frozen in liquid nitrogen and stored at −80 °C. Patient demographics were recorded at the time of tissue collection. Details of Non-Failing heart samples (N1, N2, N3, N4, N5): Sex ratio, M:F = 1:4; age 18–68 with median 62; heart EF (ejection fraction) 55–70% with median 65%; myocardial wall thickness 8–15 mm with median 13 mm. Details of Failing heart Samples (F1, F2, F3, F4, F5): Sex ratio, M:F = 3:2; age 38–67 with median 49; heart EF (ejection fraction) 15–55% with median 25%; myocardial wall thickness 6–12 mm with median 8 mm. For the failing heart: % with LVAD (2/5), ACEI/ARB (3/5), BB (2/5), and inotrope (1/5).

**Heart-muscle tissue lysate preparation for proteasomes and intracellular peptides.** The frozen heart-muscle tissue (~0.5 g) was resuspended with 300 μL of chilled PBS containing 1x protease inhibitor cocktail and was quickly homogenized into suspension. The tissue suspension was divided into two halves (for proteasome

preparation and for intracellular peptide isolation). For proteasome lysate preparation, 1 mL of chilled ATP buffer was added and lysed with glass beads and vortexed at 4 °C. After high-speed centrifugation ($16,000 \times g$) for 30 min, the clear supernatant was collected, flash-frozen in liquid nitrogen, and stored at –80 °C. For intracellular peptide isolation, 1 mL of 80 °C hot water was added to the tissue suspension and processed as described above.

### Proteasome enrichment and proteasome-trapped peptides (PTPs) isolation.
The native proteasome lysates were subjected to ultra-centrifugation at $100,000 \times g$ for 45 min at 4 °C. The first ribosome pellet was discarded, and the supernatant was further centrifuged at $150,000 \times g$ for 4 h at 4 °C. Then, the pellet was collected, washed, and resuspended with ATP buffer. For PTPs isolation, 80 °C hot water was added to the enriched proteasome suspension, incubated at 80 °C for 20 min, and processed as described in the "Intracellular peptide isolation" section.

### Cell proliferation and survival assay.
A total of 5000 HeLa cells were seeded in each well of a 96-well plate and grown either under hypoxia (1% O₂) and normoxia. After 24 h of seeding, the cells were treated with puromycin at different concentrations (250–1000 ng/mL) for different time periods (0–60 h). At each time point, 100 μg of MTT was added per well and 6 h later, 10% SDS (in 0.01 N HCl) solution was added to dissolve the crystals; 16 h after adding the SDS, the colorimetric reading was taken at 570 and 690 nm. Similarly, doxycycline-treated and untreated PSMD2-KD-inducible HEK293 cells were seeded at concentration (5000 cells/well) assay was performed as HeLa cells.

### In vivo protein synthesis assay.
HeLa or HEK293T cells were grown in usual full DMEM media followed by two quick washes with DMEM –Met and –Cys. Then, cells were incubated with DMEM containing 0.2 mM L-azidohomoalnine (AHA; Click-iT/Invitrogen), 0.5 mM L-cysteine (Sigma) and 10% FBS for various time periods (1, 2, 3 h). Subsequently, the cell lysates were prepared in 0.5% NP-40 lysis buffer (50 mM Tris 7.4 pH, 150 mM NaCl, 0.5% NP-40, 1x protease inhibitor cocktail). In total, 200 μg of total cell lysates were subjected to TAMRA-alkyne labeling (Click-iT/Invitrogen) with Click-iT Protein reaction buffer according to the manufacturer's protocol; 20 μg of TAMRA-labeled lysates were resolved in SDS-PAGE and observed in Gel-Doc system attached to the wavelength of ~555/580 nm.

### RNA isolation and real-time PCR.
Total RNA was isolated from either HeLa or HEK293T cells by using TriZol-reagent and the manufacturer protocol (Invitrogen). One microgram of total RNA was converted to cDNA by using reverse transcriptase (RT)-PCR according to the manufacturer's protocol (qPCRBIO cDNA Synthesis Kit; qPCR Biosystems). Real-time RT-PCR was performed using qPCRBIO SyGreen Blue Mix (qPCR FBiosystems) to detect mRNA expression of ubiquitin genes. The list of primers is provided in Supplementary Table 2. The real-time PCR data was analyzed using QuantStudio™ Design and Analysis Software (Thermo Fisher).

### Cryo-EM sample preparation and data collection.
Human 20S proteasomes were mixed with MonoUb-Cyclin B1-NT at a molar ratio of 1:5, incubated on ice, and frozen within 5 min. A volume of 2 μL sample was placed onto a glow discharged Quantifoil holey carbon grid (R2/1, 200 mesh, Quantifoil Micro Tools), which was freshly coated with graphene oxide (Sigma Aldrich). The grid was blotted with Vitrobot Mark IV (FEI company), and flash-frozen in liquid ethane. The 20S sample was handled similarly without added substrate and incubation. Images were taken by using a Titan Krios transmission electron microscope (Thermo Fisher) operated at 300 kV and equipped with a Cs corrector (Supplementary Fig. 6a and 7a). Images were collected using a K2 Summit direct electron detector (Gatan) in super-resolution mode at a nominal magnification of 18,000X, yielding a pixel size of 1.318 Å after two times binning (Supplementary Fig. 6 and Supplementary Table 3). Each movie was dose-fractioned into 38 frames with a dose rate of 8 e⁻/pixel/s on the detector. The exposure time was 7.6 s with 0.2 s for each frame, generating a total dose of 38 e⁻/Å². Defocus value varies from −0.9 to −1.8 μm. All of the images were collected using the SerialEM automated data collection software package[74].

### Single-particle cryo-EM data processing.
Overall, 3125 movies for the 20S + MonoUb-Cyclin B1-NT sample and 748 movies for the 20S-alone sample were collected for structural determination. Single-particle analysis was executed in RELION 3.0. All images were aligned and summed using MotionCor2 software[75]. After CTF parameter determination using CTFFIND4[76] and Gctf[77], particle auto-picking, manual particle checking, and reference-free 2D classification, particles went through a 3D classification with C1 symmetry applied, using a 20S map from previous work as initial reference[78]. Here, the number of classes for the first round of 3D classification used in the 20S-alone dataset was 5, whereas it was 8 in 20S + MonoUb-Cyclin B1-NT because of its larger dataset size. The groups with rational 20S characteristics and good resolution accounted for the major percentage (80–90%) were remained for further process. Following multiple rounds of reference-free 2D classification and 3D classification, 282,834 particles for 20S +

MonoUb-Cyclin B1-NT and 154,436 particles for 20S alone remained for further processing. After applying CTF refinement and Bayesian polishing to the remaining particles, they were auto-refined without imposing any symmetry and reached 3.38 Å resolution for 20S + MonoUb-Cyclin B1-NT and 3.22 Å resolution for 20S alone after post processing.

A further 3D classification into five classes was performed for each of the two data sets. In the 20S + MonoUb-Cyclin B1-NT dataset, class 1 with an obviously asymmetric gate configuration (in magenta, 29.1% of the population) is distinct from the other classes (Supplementary Fig. 7b). We then performed two more rounds of 2D classification on class 1 data to discard misaligned or broken particles. Further refinement on these cleaned-up particles led to a map at 4.47 Å resolution (denoted as S1; Supplementary Fig. 7c, d). For the other three classes with reasonably good structural features (2, 4, and 5), since they all appear symmetrically closed in the two rings and show highly similar features, their particles were combined and refined into a map at 3.88 Å resolution (denoted as S2; Supplementary Fig. 7d) with a normal 2S configuration. For the 20S-alone dataset, further classification did not yield significant differences between classes for the four classes showing reasonably good structural features (1–4, Supplementary Fig. 6), and their two gates all appeared symmetrically closed. For further structural analysis, we used the reconstructions before postprocessing, since these maps present better structural feature connectivity (Supplementary Fig. 7e).

### Pseudo-atomic model building and structural analysis.
The available atomic model of the human 20S proteasome (pdb: 6rgq)[79] was used as an initial model. We performed further flexible fitting of the model against the corresponding map by utilizing Rosetta[80]. The generated model was further refined using the real-space refinement function in PHENIX[81], and manually modified using COOT[82]. UCSF Chimera and ChimeraX were used for figure generation[83–85]. For 20S + MonoUb-CyclinB11-NT, the difference map was generated by using a 180° rotated (around the pseudo twofold axis) 20S + MonoUb-CyclinB11-NT map minus its original unrotated map. Program proc3d in EMAN1.9 was used for differential map generation[85]. For the difference map of the 20S alone, a similar procedure was adopted. 3D variability analyses (3DVA) analysis was conducted in CryoSparc.

### LC-MS/MS.
Samples of in vitro cleavage reactions (except those used for the identification of cyclin B1 peptides with ubiquitin remnants) were analyzed using Orbitrap Fusion Tribrid (Thermo Scientific) coupled to Ultimate 3000 Nano Systems (Thermo Scientific) following loading via an Acclaim PepMap 100 trap column (100 μm × 2 cm, nanoViper, C18, 5 μm, 100 Å; Thermo Scientific, Waltham, MA) onto an Acclaim PepMap RSLC analytical column (75 μm × 50 cm, nanoViper, C18, 2 μm, 100 Å; Thermo Scientific). The peptides were eluted with a linear 30 min gradient of 6–30%, followed by a 3 min gradient of 30–34%, 5 min gradient of 34–76%, and 10 min wash at 76% acetonitrile with 0.1% formic acid in water (at flow rates of 0.2 μL/min). These MS analyses were done in positive mode using high energy collisional dissociation (HCD) when each cycle was set to a fixed cycle time of 2 s and consisted of an Orbitrap full ms 1 scan (range: 375–1575 m/z) at 60,000 resolution with AGC target: 1e6; maximum IT: 118 ms, followed 18 Orbitrap ms2 scans (range 140–2000) at 60,000 resolution: with AGC target: 4e5; maximum IT: 118 ms; isolation window: 1.4 m/z; HCD Collision Energy: 32%. Dynamic exclusion was set to 10 s and the "exclude isotopes" option was activated.

The rest of the peptidomics samples were analyzed using Q-Exacitive-Plus or Q-Exactive HF mass spectrometer (Thermo Fisher) coupled to nano HPLC. The peptides were resolved by reverse-phase chromatography on 0.075 × 180 mm fused silica capillaries (J&W) packed with Reprosil reversed-phase material (Dr. Maisch; GmbH, Germany). The peptides were eluted with a linear 60 min gradient of 5–28%, followed by a 15 min gradient of 28–95%, and a 10 min wash at 95% acetonitrile with 0.1% formic acid in water (at flow rates of 0.15 μL/min). MS analysis by Q-Exactive HF-X mass spectrometer (Thermo Fisher Scientific) was in positive mode using a range of m/z 300–1800, resolution 60,000 for MS1 and 15,000 for MS2, using, repetitively, full MS scan followed by HCD of the 18 most dominant ions selected from the first MS scan.

### Mass spectrometry data analysis.
MS data analysis was performed with either MaxQuant version 1.6.7.0[86] or the Trans Proteomic Pipeline (TPP) v5.2.0 Flammagenitus[87]. MaxQuant searches were performed using unspecific digestion mode with a minimal peptide length of 7 amino acids. Search criteria included oxidation of methionine and protein N-terminal acetylation as variable modifications. All other parameters were set as the default. The raw files were searched against the Homo sapiens UniProt fasta database (November 2017; 20,239 sequences) supplemented with the sequences of the fusion protein of Cyclin B1-NT (1–88 aa residue) with NS3-protease. Candidates were filtered to obtain FDR of 1% at the peptide and the protein levels. No filter was applied to the number of peptides per protein. For quantification, the match between runs modules of MaxQuant was used, and the label-free quantification (LFQ) normalization method was enabled.

TPP searches were performed following RAW files conversion to mzML using MSConvert (ver 3.01157) with centroid and compressing peak list option. Searches were performed using Comet (2017.01 rev. 1) and high-resolution settings. They included 'no enzyme' cleavage specificity and oxidation of methionine and protein

N-terminal acetylation as variable modifications. All searches were conducted against protein sequences downloaded from UniProt[88]. The searches of in vitro cleavage reactions of cyclin B1 were done with a database composed of all Uniport Human proteome (November 2017; 6721 sequences), all of Human proteasome-related proteins (based on "Proteasome" Keyword in Uniprot, November 2017; 169 sequence), the sequences of modified cyclin B1, FLAG-tagged ubiquitin and Myc-tagged ubiquitin, and the sequence of the cRAP contaminant database (cRAP database released Feb 2012; 115 entries). All these sequences were supplemented with decoy sequences. Searches of other experiments were done against the *Homo sapiens* UniProt fasta database (November 2017; 20,239 sequences) supplemented with the sequences of the fusion protein of Cyclin B1-NT (1–88 aa residue) with NS3-protease and the sequence of the cRAP contaminant database (cRAP database released Feb 2012; 115 entries). PeptideProphet was used to curate peptide-spectrum matches of FDR ≤ 1% and assign representative proteins/protein isoforms. We note that this approach does not unambiguously identify peptides shorter than 6–7 amino acids. Label-free quantification of ubiquitin peptides was performed using Skyline version 4.2[89]. Peptide library was constructed based on database search results obtained for heart failure samples searched by MaxQuant as described above. The RAW files of these samples were used to calculate the sum of peak area for all peptides originated from ubiquitin.

**Identification of cyclin peptides with ubiquitin remnants**. Identification of cyclin peptides that contain ubiquitin remnants was performed by a series of separate searches using MSFragger version 3.1.1[90] via FregPipe version 14.0 (https://fragpipe.nesvilab.org/). Each of these searches was conducted using MSFragger's "non-specific peptidome" configuration against the same database used for the other in vitro cleavage reactions analyses. The search parameters for these searches are listed in Supplementary Table 4. Peptide validation for each modification (1% FDR) was done following PeptideProphet[91] and iProphet[92] analyses. The modifications and their localization were validated using PTMProphet[93]. The results obtained from the search with no modification on lysines were used to generate the sequence coverage of FLAG-ubiquitin and modified cyclin B1. The identification of ubiquitin remnants was validated following an open search with MSFragger with the same settings and by MaxQaunt search using non-specific digestion with dependent peptides option turn on.

**Cyclin B1-NT cleavage site (P1) calculation**. Peptides generated from both unmodified Cyclin B1-NT and TetraUb-Cyclin B1-NT in vitro by 20S or 26S proteasomes were searched by the TPP software and assigned PSMs (MS/MS count) to each peptide. Only peptides that were identified two out of three replicates were considered for further analysis. P1 positions at cleavage sites were assigned with a value calculated from the integrated MS/MS count of all the corresponding peptides. Next, the relative cleavage preference for the 20S over the 26S proteasome at each P1 position was calculated and represented as $\log_2$ of the corresponding ratios. P1 positions with a zero value were considered as 0.1 for the relative cleavage preference calculation. In the case of in vivo P1 position relative cleavage preference calculation, a similar method was used with a modification that considered peptides appearing even in one replicate from independent intracellular peptidomic analysis repeats.

**Bioinformatics analysis of MS/MS data and statistics**. The sample clustering and heatmap analysis of unmodified and ubiquitinated Cyclin B1-NT in vitro cleavage experiment was performed as follows: The PSM counts per peptide were converted into a sample-by-peptide matrix and samples with less than 50 PSMs were discarded, along with peptides observed only in single samples. The distances between the samples were obtained by performing MDS analysis as implemented in the plotMDS function from edgeR v. 3.26[94]. The distances from resulting ordinations were utilized to cluster the samples using the Ward's method. The heatmap was obtained by normalizing the PSM counts as counts-per-million and rescaling them to the maximum PSM counts per peptide. The maxima were plotted alongside the heatmap. Analogous analyses were performed for P1 positions instead of peptides. Cleavage site amino acid composition preferences were analyzed by building information-content sequence logos and differential sequence logos using dagLogo v. 1.22.2 and DiffLogo v. 2.8.0, respectively[95]. For the heart failure peptidomics experiment, the PSMs were obtained as described above. Ordination of samples in MDS axes was performed with edgeR and was based on protein-level PSM counts. Differences between size distributions of cleavage peptides in multifactorial experimental designs were analyzed by fitting Poisson-family GLMs with lme4 v. 1.1-[96], inspecting significant model terms with ANOVA, and running post hoc tests with emmeans v.1.3.5. Otherwise, pairwise differences were tested with the Mann–Whitney U test. Most of the data manipulation steps, diagrams, and statistical tests were performed in R v.3.6.1. The prediction of protein disorder was performed using IUPRED2A[97] (https://iupred2a.elte.hu/). The sequences of relevant proteins were downloaded from Uniport and the resulting FATSA file was submitted prediction with IUPRED2 using the default settings. The Disorder score for each protein was set as the average value of IUPRED score of its amino acids (calculated by in-house script). Boxplot graphs were generated using BoxPlotR[98] (http://shiny.chemgrid.org/boxplotr/) or Graphpad Prism V5.

**Quantification of gels/blots, statistics, and reproducibility**. ImageJ software (Mac Biophotonics) was used for the quantification of bands generated in Coomassie-stained SDS-PAGE and immunoblots. Unpaired *t*-test was performed with 95% confidence intervals using Graphpad Prism V5 software for specific experiments, as indicated in the figure legends. All experiments in the study are executed and tested for their reproducibility at least three times independently. Most of the cell biology experiments are performed with minimum of three biological repeats. All the biochemical experiments shown here by representative images are repeated independently at least in triplicates.

**Reporting summary**. Further information on research design is available in the Nature Research Reporting Summary linked to this article.

## Data availability
All analyzed data are uploaded along with this manuscript as supplementary data files and source data (uncropped images and datapoints underlying in graphs). The mass spectrometry proteomics data have been deposited to the ProteomeXchange Consortium via the PRIDE partner repository with the dataset identifier PXD018711, PXD018722 and PXD028912. Cryo-EM maps determined from the 20S-alone dataset has been deposited at the Electron Microscopy Data Bank with accession codes EMD-13389 (before post processing) and EMD-31730 (after post processing by RELION), and the associated atomic model has been deposited in the Protein Data Bank with accession code 7PG9. Cryo-EM maps determined from the 20S + monoUb-Cyclin B1-NT dataset have been deposited at the Electron Microscopy Data Bank with accession codes of EMD-31728 (S0), EMD-31724 (S1), and EMD-31727 (S2), and related models have been deposited in the Protein Data Bank under accession code of 7V5G (S1) and 7V5M (S2). S0 was further 3D classified into S1 and S2 for analysis; therefore there is no PDB entry corresponding to EMD-31728 (S0). The human 20S proteasome structure 6RGQ was used as a reference model for our study. Source data are provided with this paper.

## Materials availability
Biological materials (Plasmids) generated in this study will be readily available from the corresponding author.

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

## Acknowledgements

We thank Peter Tsvetkov (Broad Institute of MIT and Harvard) for sharing Cell lines and shRNA constructs, Simone Engelender for the α-synuclein construct, Noa Reis for critical advice and aid in cloning, Inbar Magid Gold for recombinant enzymes, Tehila Bar Kafra for coding MS analysis software, Nitzan Dahan for aiding with the FACS and microscopy analyses, Ilana Navon and Tamar Ziv and rest of the team of Smoler Proteomics Facility for performing MS analyses, and Cheng Huang and Ralf Schittenhelm from Monash Biomedical Proteomics Facility for performing MS analyses. This research is supported in part by Israel Science Foundation grants (755/19 to M.H.G.; 179/15 to A.B.; 1623/17 and 2167/17 to O.K.). NSFC-ISF (2512/18 to M.H.G. and 3181143028 to Y.C.), NSF-BSF (2017727 to M.H.G.), China National Postdoctoral Program for Innovative Talents (BX2021310 to C.X.), and MSCA-IF Horizon-2020 (748804 to I.S.). This project also has received funding from the European Research Council (ERC) under the European Union's Horizon-2020 research and innovation program (grant agreement No. 831783 to A.B.). PS is a Rappaport Family Technion Integrated Cancer Center (R-TICC) fellow, A.B. is the Jordan and Irene Tark Academic Chair, M.H.G. is the Israel Isaac and Natalia Kudish Academic Chair.

## Author contributions

I.S., Y.C., O.K., A.B. and M.H.G. designed the research and developed the project. I.S. and P.S. conducted cell biology experiments. I.S. and R.M. conducted cloning and biochemical experiments. S.M.M. and S.K.S. performed chemical synthesis of proteins. M.P.S. carried out proteomic sample preparation. A.R. and O.K. analyzed proteomic data. C.X., Z.D., Y.W., and Y.C. performed cryo-EM analysis. S.D. procured human heart tissue. I.S., O.K., A.B. and M.H.G. wrote the first version of the manuscript. All authors read and commented on the manuscript.

## Competing interests

The authors declare no competing interests.
