## [Peer Review File · Nature Communications]

REVIEWER COMMENTS

Reviewer #1 (Remarks to the Author):

Sahu et al. report a curious finding in which they observed that a ubiquitin tagged unstructured model substrate can be degraded by the 20S proteasome including its ubiquitin tag. The authors present a huge amount of data supporting their observation and even find natural conditions in which this activity might be of relevance in cells in their native environment. As this activity has not been seen before and is an exciting finding it is definitely worth publishing. I do believe the main claims are supported by the data presented, however, I see quite few flaws in the presentation of the manuscript and some of the data analysis and would recommend revising the manuscript before publishing.

1) My main issue with the manuscript is the overall presentation and the logic of how it is written. It is full of grammar and style issues. I would suggest considering restructuring and rewriting.

The manuscript nicely starts with an in depth in-vitro analysis that defines activities that the authors suggest to be signature activities of the 20S proteasome. While this presentation is solid, the mechanistic investigation of these partially really surprising signatures is not and many questions are only scratched on the surface. (see my point 3). Next the authors design an in-vivo system using an artificial substrate and a knockdown background, that nicely recapitulates the in-vitro data. However, in the middle of it they jump into Hypoxia conditions and then continue to jump between the different in-vivo systems which makes the rest of the manuscript harder to follow. To improve that I would give two different recommendations:

A) At the very least the two in-vivo systems should be disentangled and first the results from the artificial cells should be shown and hypoxia should be the ultimate part of the manuscript.

B) Alternatively given the huge amount of data it would be probably justified to split the manuscript into two and have one for the signature activities of the 20S proteasome and one for the effects in hypoxia. This would give room for making the first paper more molecular mechanistic.

In General, the manuscript would benefit from having less experiments but with more focus.

2) Also on a smaller scale within paragraphs the readability should be improved. Often the rationale or the approach of an experiment is explained after the results. (e.g. pg9 Pyr-41 is introduced as several lines after the result of it was mentioned, or pg 11 "synthesis arrest" comes out of nowhere.)

I would suggest writing in a more comprehensible way in which every paragraph starts with rational including the approach of an experiment followed by the result.

3) While I am intrigued by the finding that the 20S proteasome can degrade ubiquitinated conjugates and the authors provide a lot of data to support this observation I am lacking a mechanistic explanation of how this is possible. Ubiquitin is a particularly good folder and it seems very unlikely that it would be simply unfolded without any extra effort especially as pointed out by the authors that other folded domains are not able to be degraded by the 20S proteasome alone when fused to an unstructured protein.

As this is indeed the most important finding I would like to ask the authors to support it with more experiments that elucidate what the mechanism is that allows this. As I will outline in my next point the cryo EM analysis is flawed and does not provided enough evidence for the possibility of a branched peptide to enter into the proteolytic chamber I would suggest the following experiments but I would be happy to see anything else explaining the curious finding and disregarding trace contaminations that do that job.

A) The authors constructed a very elaborately ubiquitinated substrate but hardly use all the different labels in their experiments. More consequent Western blotting of all three labels would make everything more believable. Also an additional C-terminal tag to the substrate would actually enable the authors to monitor the degradation better and support their finding.

B) If the rather complicated lysine linked ubiquitin is co-degraded I would assume it should be no issue to degrade the linear fusion of ubiquitin with the substrate.

C) If the observation is true the degradation should happen ATP-independent. So, I would wonder what would happen to the apyrase treated 26S purification, that would substantially just free 20S particles.

D) A crosslink within the ubiquitin should prevent the degradation.

E) The peptides of that contain the linkages have not been found. However, one could try to enrich for those as their existence in the solution would be critical evidence.

F) Can something else be linked via the lysin and is also degraded?

4) The cryo EM analysis has a few flaws and should be overworked. The EM analysis is the main support for mechanistic explanation as summarized in Fig 7. Thus this needs to be properly validated. I am certainly not convinced that the gate is indeed open as described by the authors

I do thank the authors for providing maps and models. But by looking at the maps it seems like that the substrate incubated map shows anisotropic resolution in the direction of the cylindrical axis. This is very evident when looking on the map from the side seeing multiple structural features smear out in this direction. I would thus assume that the angular distribution between the two maps is very different and thus the two maps appear different while they might not be. From looking at the map I

would expect that there are way more top views in the substrate bound map than in the apo map. Please provide an angular distribution plot for both datasets to clarify.

I would also suggest to make a somewhat more modern analysis to understand the differences. First of all, to minimize bias in the two datasets I would suggest to mix the two datasets together analyse and classify them together and then see afterwards how the distribution over different conformers looks like. This would remove any bias from the analysis.

Additionally, several approaches to disentangle different continuous conformational changes have been published recently and I would recommend to use cryodrgn or cryosparc's 3D variability analysis to get a more clarified picture. Also to give the cryo EM analysis more meaning one would need to find at least smeary density for the ubiquitin.

However, I believe if the authors want to make a point of the cryo EM analysis it should be part of the main manuscript.

5) Finally a few general technical comments:

A) In Western Blots there is no linear relationship between signal intensity and protein amount over a wide range. Thus, protein amounts cannot be used synonymously with the band intensities. Thus data should be labeled as relative band intensities and not “% of substrate left or proteasome content”. This would need a proper calibration, which I assume was not done.

B) Boxplots require a sample size of at least 5 data points (Krzywinski, M., & Altman, N. (2014). Visualizing samples with box plots. *Nature Methods*, 11(2), 119–120. <https://doi.org/10.1038/nmeth.2813>)

C) T-test on frequencies (proportions) is highly debatable. I would strongly advice against it as frequencies are not normal distributed. While it seems unfortunately almost the standard in biology a population size of three is hardly sufficient for a t-test as well

Minor issues:

1) The writing of the manuscript has a few issues especially regarding grammar. For instance, articles are rarely used. To name one example, it should be degradation by “the”26S proteasome and not just by 26S proteasome. I would recommend letting the manuscript being edited by a native English speaker to avoid grammar mistakes. Also paragraphs contain casual language that should be made more precise.

2) Cryo EM does not produce an “electron density”. So please don't refer to it (p.8)

3) The statement that the purified 26S proteasomes don't contain associated DUBs is slightly misleading as they of course must contain rpn11 which is an intrinsic DUB. Also Extended data Figure only checks for the presence of USP14 but there are more proteasome associated DUBs known.

4) The presence of supplementary and extended data Figures at the same time is very confusing and there should be only one kind of supplementary material.

5) The paper contains mixed nomenclature for proteasome subunits. E.g. the name PSMD2 is used alongside Rpn10. As the proteasome nomenclature is confusing enough I would suggest to stick either to the PSM or the Rpn nomenclature and don't mix both.

6) Ext. Fig. 2: Figure legend is wrong explained. Both substrates are in a and b. But the difference between a and b is the 20S or the 26S proteasome.

7) Sup. Fig. 1b: One grey dot has more than 2-fold preference for the 26S proteasome (between P1=30-40).

8) Fig. 3a: The Bortezomib control seems somewhat random and half-hearted. If used it should be used in both circumstances.

9) Fig. 3a,b: bands for tetra Ub and tetUb-CycB1 in a have about 10 kDa more than in b.

10) Fig. 3b: add an anti-HA WB to see whether deubiquitinated CycB1 accumulates.

11) Fig. 3c: supplement anti-Ub/anti-HA WB.

12) Sup. Fig. 2: include some detail and rationale about the used inhibitors. (IAN, OPA)

13) Ext. Fig. 2d in text = Ext. Fig. 3d.

14) Ext. Fig. 4b: The figure indicates that a "Poly-ubiquitin" AB was used for the WB. Wherefore were HA-CycB1-NT expressing HEK293T cells used, when the entirety of ubiquitinated proteins are monitored?

15) Ext. Fig. 5a: show the whole blot. The CycB1 band could decrease due to ubiquitination.

16) Ext. Fig. 5g: show the whole blot. For WT CycB1, only non-ubiquitinated band is shown.

17) Ext. Fig. 5h: Why is KO CycB1 present as multiple bands?

18) Fig. 4b,g: n=3, which is too low for a box plot and questionable for statistical tests.

19) Fig. 4b,d: Data derives from immunoblot intensities.

- 20) Fig. 4c: Only non-ubiquitylated CycB1 is shown. Show the whole blot.
- 21) Ext. Fig. 6d: Show the whole blot to see if poly-Ub-alpha-Syn accumulates.
- 22) Ext. Fig. 7b: Data derive from WB band quantification.
- 23) Fig. 4e: Show the whole blot to see ubiquitinated CycB1.
- 24) Sup. Fig. 6a: "Pure 26S" contains some PA200 and "Pure 20S" contains some PA28alpha. Supplement PA28beta/gamma IB.
- 25) Fig 4g: Which test was used? T-test probably not valid here.
- 26) Fig. 4h: The data do not really seem significant.
- 27) Sup. Fig. 6b,c,d in text = Sup. Fig. 7b,c,d.
- 28) Fig. 5a: show 0h CHX chase of Hi20S.
- 29) Ext. Fig. 7b in text: The figure shows increased abundance of 20S during hypoxia, not enhanced function.
- 30) Fig. 5c: Data derive from WB => No linear relationship.

Reviewer #2 (Remarks to the Author):

The paper of Sahu et al. describes a new and astonishing feature of the 20S proteasome – the degradation of ubiquitinated, unfolded proteins.

It is for some years accepted by now, that the 20S proteasome has a distinctive function in eukaryotic cells, especially under stress conditions. However, whether unfolded, ubiquitinated proteins are degraded by the 20S or the 26/30S form of the proteasome was unknown. Therefore, this manuscript shows that both proteasomal forms contribute and that under proteotoxic stress condition the 20S proteasome is the most active form, as was shown earlier for oxidative stress conditions. However, the finding that the 20S is also able to degrade ubiquitinated proteins is new.

The experiments are well designed and performed. However, a few additional questions should be answered:

1. A bit puzzling is the finding that the 20S is not able to degrade Ub or the Ub-tetramer in its free form. Is the Ub-tetramer not properly folded if attached to the CyclinB1-NT? Or is the CyclinB1-NT just opening the gate? Is a proteasome with an open gate able to degrade Ub and the Ub-tetramer?

2. The authors analyzed the peptide products of the 20S degradation. How does the peptide with the Lys-Ub-linkage look like? Is there any consistent distance between the last peptide bond cleaved and the linkage (N- and C- terminally and in the Ub.chain). Here the products should be characterized in detail. The same is true for the degradation within the Ub-tetramer.

Reviewer #3 (Remarks to the Author):

In their manuscript „20S as a stand-alone proteasome in cells can degrade the ubiquitin-tag” Sahu et al. investigate how the 20S catalytic complex differs in its function when it is free or part of the intact 26S proteasome.

The authors use state of the art chemical synthesis to generate a set of site-specifically ubiquitylated proteins that can be used as substrates for both 20S and 26S complexes in order to specifically probe their respective roles in protein degradation in vitro. By combining their approach with various mass-spectrometry based readouts the authors are able to dissect specific roles for both 20S and 26S complexes in substrate selection and peptide generation and demonstrate a novel feature of the 20S complex; its ability to degrade the ubiquitin tag along with the target protein.

While it has been shown before that the free 20S complex exerts specific catalytic functions, this clever use of chemical synthesis puts the authors into the unique position to assign function to specific ubiquitylation states of substrate proteins, which clearly both broadens and deepens our understanding of 20S catalytic function.

The authors then go on and show a potential cellular role of the 20S complex in hypoxia and human failing-heart cells and suggest that elevated roles of 20S heighten cell survival under these conditions, which are both of great interest.

Thus, this manuscript provides an extremely timely and valuable resource for those who study proteasome-mediated protein degradation and should make for a great fit for Nature Communications.

However, some points need clarification.

While the various MS-based workflows appear to be generally solid and executed carefully, some explanation on how intracellular peptidomics was carried out is missing. The experimental side is

somewhat explained in the methods section and the capture to Figure S7, but the MS and downstream analysis side needs additional explanation.

Regarding the specific degradation of ubiquitylated samples by 20S/26S (extended Data Fig. 2).

The data shows that the endpoints of the enzymatic digestions of both mono and di-ubiquitylated CyclinB1 are identical and only slightly differ in their respective rates; moreover, the rates for mono and di-ubiquitylated cyclin B1 appear to be inverted. It would be great if the authors could elaborate on why they think this is the case.

Fig. 5: 20S proteasome contributes to degradation of ubiquitin conjugates in vivo:

How exactly was the number of proteasomes determined? More importantly, how the number of ubiquitin-derived peptides? Maybe the authors could clarify this.

The authors make some suggestions on why proteolysis of the ubiquitin tag may be a feature of the 20S complex. Could the same goals not be reached without digestion of the tag? -perhaps the authors could speculate a bit further.

Reviewer #4 (Remarks to the Author):

This work by Sahu et al. is a tour-de-force investigation of the novel function of 20S proteasome. Although the role of 26S proteasome in protein degradation has been well established, the potential role of the 20S proteasome was not clear. This study addressed this intriguing question through reconstituted proteasome system combined with well-defined, synthesized polyUb cyclin B1 (N-terminal disordered region, CyclinB1-NT). The results were further corroborated using cultured cells and human heart myocardial tissues. The results obtained revealed interesting distinctions between the 26S and 20S proteasomes in degrading unmodified and ubiquitinated proteins, particularly disordered proteins as represented by CyclinB1-NT. This study is of high significance and suitable for publication in Nature Communications. Below are questions that need to be addressed to further improve an otherwise already strong manuscript.

The 26S proteasome prep used in this study lacked the proteasome-associated DUBs. This affects the protein degradation outcome in terms of rate and peptide distribution. Can the authors do the same

experiment comparing a 26S proteasome with DUBs associated? This is an important experiment for comparison with the degradation study reported in cells in which DUBs are associated with proteasomes.

In Fig. 2h there are clear gaps for mono- and tetraUb-CyclinB1-NT, but not in unmodified CyclinB1. What are these regions and why such a difference exists? Please add residue numbers to the graph.

Fig. 3d, although Ub peptides were detected by MS/MS, they cannot be attributed specifically to one of the four Ubs except the Ub peptides containing the tag sequences. The figure is misleading and needs to be modified. In Fig. 3c, the mono and diUb-CyclinB1-NT band should be indicated.

The cellular degradation study used CyclinB1-NT fused to NS3 protease as stated in the text. The gel analysis results shown in Fig. 4 seems to suggest that the CyclinB1-NT was detected as a stand-alone protein. Was this due to cleavage of NS3 from the fusion protein? Please clarify. This is important for the correct interpretation of the results.

The degradation of tetraUb-CyclinB1-NT by 20S was much slower (~ 10 fold, with a half-life of 200 mins) than unmodified CyclinB1-NT as shown in the in vitro system. Coupled with the cryo-EM results presented in the manuscript it is reasonable to suggest that the ubiquitinated substrate is bound to the 20S proteasome normally but its proteolysis is much slowed down. Alternatively, the slower processing of polyUb-CyclinB1-NT may be due to less efficient recruitment of the protein to the 20S proteasome. Have these possibilities been addressed? The authors may do a 20S proteasome assay with an equal mixture of unmodified CyclinB1-NT, mono-, di- and tetraUb CyclinB1-NT and see how they are degraded. If the latter is true unmodified CyclinB1-NT is likely degraded first. Otherwise, an inhibition may be observed due to the presence of polyUb-CyclinB1-NT.

Would elevated level of 20S present an issue given that many fully or partially disordered proteins exist in human cells? How is the 20S proteasome's activity regulated to avoid toxic effect to cells? This should be considered in the discussion.

Following are all reviewers' comments and suggestions followed by our response or answer. (The highlighted parts are the corresponding changes made in the main text.)

REVIEWER COMMENTS

Reviewer #1 (Remarks to the Author):

Sahu et al. report a curious finding in which they observed that a ubiquitin tagged unstructured model substrate can be degraded by the 20S proteasome including its ubiquitin tag. The authors present a huge amount of data supporting their observation and even find natural conditions in which this activity might be of relevance in cells in their native environment. As this activity has not been seen before and is an exciting finding it is definitely worth publishing. I do believe the main claims are supported by the data presented, however, I see quite few flaws in the presentation of the manuscript and some of the data analysis and would recommend revising the manuscript before publishing.

1) My main issue with the manuscript is the overall presentation and the logic of how it is written. It is full of grammar and style issues.

We apologize for that. We have rewritten the entire text and polished language and style, including having it read and checked by unrelated readers. In the process we also restructured the entire flow of the text, starting with the *in vitro* mechanistic characterization, followed by the *in vivo* validation, and ending with evidence for the possible role of the 20S in cells. We attach a text file with all changes highlighted as track changes for your convenience.

I would suggest considering restructuring and rewriting.

The manuscript nicely starts with an in depth *in-vitro* analysis that defines activities that the authors suggest to be signature activities of the 20S proteasome. While this presentation is solid, the mechanistic investigation of these partially really surprising signatures is not and many questions are only scratched on the surface. (see my point 3). Next the authors design an *in-vivo* system using an artificial substrate and a knockdown background, that nicely recapitulates the *in-vitro* data. However, in the middle of it they jump into Hypoxia conditions and then continue to jump between the different *in-vivo* systems which makes the rest of the manuscript harder to follow. To improve that I would give two different recommendations:

A) At the very least the two *in-vivo* systems should be disentangled and first the results from the artificial cells should be shown and hypoxia should be the ultimate part of the manuscript.

B) Alternatively given the huge amount of data it would be probably justified to split the manuscript into two and have one for the signature activities of the 20S proteasome and one for the effects in hypoxia. This would give room for making the first paper more molecular mechanistic.

In General, the manuscript would benefit from having less experiments but with more focus.

Thank you for this general suggestion to improve the flow of the manuscript (clearly a result of in-depth reading and analysis of the manuscript). We chose option A. The revised manuscript has been rewritten accordingly, starting with the *in vitro* system, followed by validation in a cellular model (Hi20S) and culminating with *in vivo* observations under hypoxia and in cardiomyopathy. While doing so we also polished the grammar and tightened the text. We attach a text file with all changes highlighted as track changes for your convenience.

2) Also on a smaller scale within paragraphs the readability should be improved. Often the rationale or the approach of an experiment is explained after the results. (e.g. pg9 Pyr-41 is introduced as several lines

after the result of it was mentioned, or pg 11 “synthesis arrest” comes out of nowhere.) I would suggest writing in a more comprehensible way in which every paragraph starts with rationale including the approach of an experiment followed by the result.

In the revised manuscript we tried our best to follow this guideline. We’ve added a rationale before each experiment, and each new molecule or term is explained the first time they are mentioned. **These changes can be easily found in the track changes file attached.**

3) While I am intrigued by the finding that the 20S proteasome can degrade ubiquitinated conjugates and the authors provide a lot of data to support this observation I am lacking a mechanistic explanation of how this is possible. Ubiquitin is a particularly good folder and it seems very unlikely that it would be simply unfolded without any extra effort especially as pointed out by the authors that other folded domains are not able to be degraded by the 20S proteasome alone when fused to an unstructured protein.

As this is indeed the most important finding I would like to ask the authors to support it with more experiments that elucidate what the mechanism is that allows this. As I will outline in my next point the cryo EM analysis is flawed and does not provide enough evidence for the possibility of a branched peptide to enter into the proteolytic chamber I would suggest the following experiments but I would be happy to see anything else explaining the curious finding and disregarding trace contaminations that do that job.

A) The authors constructed a very elaborately ubiquitinated substrate but hardly use all the different labels in their experiments. More consequent Western blotting of all three labels would make everything more believable. Also an additional C-terminal tag to the substrate would actually enable the authors to monitor the degradation better and support their finding.

Initially, the rationale of painstakingly synthesizing individual ubiquitin units with different tags was to follow the fate of each ubiquitin unit during a degradation experiment. Having found that ubiquitin was not released from the substrate in a 20S reaction, made it less critical to follow all tags in each experiment. At this stage, resynthesis of the entire panel of developed substrates would be beyond a reasonable timeframe. Nevertheless, in supplementary Figure 4 (**supplementary file p. 8-9, main file p. 9**) we document the reaction progression using all tags. Moreover, those tags were helpful in identifying the source of the ubiquitin-derived peptides after proteolysis by MS/MS. For instance, chimeric peptides spanning both the tag and the N-terminus of ubiquitin were positively identified for both proximal and distal ubiquitin units and are presented in new Figure 3C (**main text p. 10 top, p. 53**). The fate of the conjugated substrate was followed by an N-terminal tag (HA), a specific antibody (N-term cyclinB1), and full coverage of its sequence by MS/MS.

p. 10: “We positively identified peptides spanning the entire polyubiquitin chain including the Myc and Flag epitopes on the proximal and distal ubiquitin units, respectively, aided by the differential tagging of the distal and proximal ubiquitin units. Chimeric (non-tryptic) peptides spanning these epitopes provided additional evidence for proteolysis of the entire TetraUb chain by 20S proteasomes (Fig. 3c and **Supplementary Datasheet S5**).

B) If the rather complicated lysine linked ubiquitin is co-degraded I would assume it should be no issue to degrade the linear fusion of ubiquitin with the substrate.

Evidence for degradation of a linearly fused ubiquitin by 26S has been published (Guterman, A. et al (2004) J Biol Chem. 279, 1729-1738). Nevertheless, the location of the ubiquitin tag likely influences the outcome and therefore we invested in attaching a K48-linked ubiquitin/ubiquitin-chains to the “natural” (i.e. published) site on the substrate (**main text p. 5 top**).

C) If the observation is true the degradation should happen ATP-independent. So, I would wonder what would happen to the apyrase treated 26S purification, that would substantially just free 20S particles.

We reiterate that all 20S complexes were purified in absence of ATP, and all 20S protease experiments were performed without any added Mg₂ATP salt (main text p. 28, p. 9 middle). Moreover, we now add a reaction in the presence of EDTA or in the presence of sodium ortho-vanadate to block any residual ATP or ATPases (supplementary Fig 4B, supp file p. 8-9).

D) A crosslink within the ubiquitin should prevent the degradation.

In order to make this experiment meaningful, we would have to crosslink ubiquitin first, and then attach it via an isopeptide bond to the substrate. This proved to be too challenging to crosslink the synthetic precursors without affecting the chemically reactive groups for the final ubiquitin conjugates (Figure 1B). We chose an alternative approach, in which we took either chemically synthesized or enzymatically conjugated ubiquitin chains without an unfolded stretch, as well as ubiquitin fused to an additional globular domain, (in the case of Di-Ubiquitin) and incubated with the 20S. The result was unequivocal: ubiquitin without an unfolded conjugate was not degraded (Supplemental Figure 5; main text p. 9 bottom, supp file p. 10).

E) The peptides of that contain the linkages have not been found. However, one could try to enrich for those as their existence in the solution would be critical evidence.

We appreciate the need for this critical evidence and thank the reviewer for bringing it up. The results included in the previous version of the manuscript were designed to compare cleavage sites between 20S and 26S and therefore reactions were deliberately terminated at early time points to decrease the potential of secondary cleavage sites. In the initial set of experiments, we did search for these branched peptides, but they were not found. In order to enrich these branched peptides, we conducted a new set of experiments terminating reactions at later time points, thereby enabling most of the substrate to be proteolyzed by 20S. Positive identification of linkages required a strategy (for MS/MS) that can deal with three degrees of freedom in “non-tryptic branched peptides”. We successfully repeated proteolysis of ubiquitin-conjugates with 20S proteasomes for longer time points and adapted the MSFragger tool for this purpose. **We are thrilled to obtain positive evidence of branched peptides generated by 20S proteolysis from an isopeptide-linked ubiquitin-conjugate.** A full description of this strategy is included in the revised manuscript with new figure panels that illustrate the complexity of products around this isopeptide linkage (new Fig. 3d-e).

This information can be found in the main text –

p. 2: “Ubiquitin remnants on branched peptide-products identified by MS/MS and flexibility in the 20S gate observed by Cryo-EM reflect the ability of the 20S to proteolyse an isopeptide-linked ubiquitin-conjugate.”

p. 10: “In order to address the possibility of a branched peptide entering into the proteolytic chamber, we performed deep analysis of all products by adapting the MSFragger tool to search for variable ubiquitin remnants on proteasome-digested cyclinB1 (Supplementary Data 6). A total of 18 different branched peptide sequences on lysine64 of cyclinB1 were identified with high confidence based on their MS/MS fragmentation spectra (Fig. 3d-e and Supplementary Table 1). This result provides direct evidence for the ability of the 20S proteasome to process a branched polypeptide, specifically in this case to cleave around an isopeptide bond.”

p. 11: “This partially “open gate” conformation is reminiscent of the activated form of the 20S CP¹⁴, and is of sufficient dimensions to enable entry of the branched polypeptide at the site of ubiquitin conjugation to CyclinB1-NT.”

p. 54: “The illustration represents the branched peptides at the isopeptide linkage of the C-terminus of ubiquitin to the K64 residue of cyclinB1 identified by non-tryptic MS/MS of reaction products. Three major types of branched peptides at K64 of CyclinB1 were detected with variants of ubiquitin remnants:

K64—GG, K64—GGR, and K64—GGRL. e, MS/MS spectra of one of the detected branched peptides.”

F) Can something else be linked via the lysin and is also degraded?

This is an interesting concept, but executing it properly would have been beyond our timeframe.

4) The cryo EM analysis has a few flaws and should be overworked. The EM analysis is the main support for mechanistic explanation as summarized in Fig 7. Thus this needs to be properly validated. I am certainly not convinced that the gate is indeed open as described by the authors

I do thank the authors for providing maps and models. But by looking at the maps it seems like that the substrate incubated map shows anisotropic resolution in the direction of the cylindrical axis. This is very evident when looking on the map from the side seeing multiple structural features smear out in this direction. I would thus assume that the angular distribution between the two maps is very different and thus the two maps appear different while they might not be. From looking at the map I would expect that there are way more top views in the substrate bound map than in the apo map. Please provide an angular distribution plot for both datasets to clarify.

I would also suggest to make a somewhat more modern analysis to understand the differences. First of all, to minimize bias in the two datasets I would suggest to mix the two datasets together analyse and classify them together and then see afterwards how the distribution over different conformers looks like. This would remove any bias from the analysis.

Additionally, several approaches to disentangle different continuous conformational changes have been published recently and I would recommend to use cryodrgn or cryosparc's 3D variability analysis to get a more clarified picture. Also to give the cryo EM analysis more meaning one would need to find at least smeary density for the ubiquitin.

We are thankful and grateful to our reviewer for these constructive suggestions. As suggested, here we provide the angular distributions of the 20S+monoUb-CyclinB1-NT and the 20S only maps (Fig. R1a and Supplementary Fig. 8a). The overall angular distributions of the two maps appear in a similar trend, i.e. both have preferred top-view distributions and lacking side views in the same direction (indicated by a dotted black circle), just with the 20S only map reconstructed from more particles (154,436 particles versus 72,086 particles, Fig. R1a). Even so, the gate of the 20S only map appears in similar closed status while that of the 20S+monoUb-CyclinB1-NT map appears asymmetric, implying that this level of preferred orientation is not the reason for the asymmetric status of the gate in the 20S+monoUb-CyclinB1-NT map.

We also followed the suggestion from the reviewer to mix the two datasets together and obtained a 3D map reconstructed from the combined data, which shows similar angular distribution (Fig. R1b and Supplementary Fig. 8b) to that of the two original datasets (Fig. R1a). We further 3D classified the combined dataset into two distinct classes (Fig. R1c-d and Supplementary Fig. 8c-d). Class 1 has 84,208 particles (37.2% of the population) and shows asymmetric α -ring gate status, with one end appearing more open than the other end (Fig. R1c), comparable with the gate density characteristics of the 20S+monoUb-CyclinB1-NT map. While class 2 shows the symmetrically closed gate as the 20S only map dose (Fig. R1d). Noteworthy, similar population (37.2%) of the particles falls into Class 1, close to the population of the 20S+monoUb-CyclinB1-NT particles in the combined dataset (72,086 particles, 31.8%); also, the angular distributions of the Classes 1/2 map are comparable with that of our 20S+monoUb-CyclinB1-NT and the 20S only maps (Fig. R1a). Collectively, these observations further substantiate the notion that the asymmetry of the gate status in the 20S+monoUb-CyclinB1-NT map is mostly like caused by the presence of substrate instead of by the preferred orientation of the particles.

Moreover, we performed the 3D variability analyses (3DVA) using CryoSparc (Punjani and Fleet, 2021). Our 3DVA analysis suggests that the patterns of the first three 3D variability components (relative to the mean density) for 20S+monoUb-CyclinB1-NT (Fig. R2 and Supplementary Fig. 9a) appear quite different

from those of the 20S only map (Fig. R3 and Supplementary Fig. 9b). This further supports our notion that the 20S only map and the 20S+monoUb-CyclinB1-NT map is in distinct states, and the 20S+monoUb-CyclinB1-NT map is more dynamic in the gate region, which might be caused by substrate association

Still, we should point out that for the substrate presented dataset, due to the compositional heterogeneity (substrate-bound/unbound) and the uncontrollable location/status of the substrate relative to the 20S, we did not fish out the density corresponding to the substrate.

Fig. R1: Angular distribution and gate status of the cryo-EM 3D reconstructions. a, Angular distribution plot of the 20S+monoUb-CyclinB1-NT and the 20S only cryo-EM maps. **b**, Angular distribution plot of the map reconstructed from the combined dataset. **c-d**, Angular distribution plot and close-up views of the two α -rings for the Class1 (**c**) and Class 2 (**d**) maps, which are 3D classified from the combined dataset and further auto-refined.

Fig.R2: Central slices of the first three 3D variability components for the 20S+monoUb-CyclinB1-NT map. Positive (red) and negative (blue) values correspond to density to be added and subtracted from the mean density, respectively.

Fig.R3: Central slices of the first three 3D variability components for the 20S only map. Positive (red) and negative (blue) values correspond to density to be added and subtracted from the mean density, respectively.

However, I believe if the authors want to make a point of the cryo EM analysis it should be part of the main manuscript.

We accepted the reviewer's suggestion and we do wish to include the CryoEM in the main text as we concur that it provides significant insight. With this validation that supports our notion that the 20S only map and the 20S+monoUb-CyclinB1-NT map are in distinct states, and with the information that the 20S+monoUb-CyclinB1-NT map is more dynamic in the gate region compared to 20S only map, which might be caused by substrate association, we have decided to include CryoEM data in main figure 4 (as well as in new supplementary figures S6-S9).

The main additions are now found in:

New main Fig. 4 p. 55.

New supplementary figures S8, S9 supplementary text p. 15-17 (in addition to the previous supplementary figures now numbered as S6-S7 p. 11-14 of the supplementary file).

Main text p. 10: “We note that the angular distribution of the two data sets was similar (Supplementary Fig. 8a). Even after mixing the two data sets, 3D classification yielded similar classes with similar angular distribution (Supplementary Fig. 8b-d).”

Main text p. 11: “3D variability analyses (3DVA) map using CryoSparc also supports the dynamic nature of the gate region of the 20S complex in the presence of monoUb-CyclinB1-NT, which might be caused by substrate association (supplementary Fig. 9).”

5) Finally a few general technical comments:

A) In Western Blots there is no linear relationship between signal intensity and protein amount over a wide range. Thus, protein amounts cannot be used synonymously with the band intensities. Thus data should be labeled as relative band intensities and not “% of substrate left or proteasome content”. This would need a proper calibration, which I assume was not done.

We have corrected the relevant figures accordingly. For instance, New Fig 2 p. 51: the Y-axis now read “Relative band intensity”.

B) Boxplots require a sample size of at least 5 data points (Krzywinski, M., & Altman, N. (2014). Visualizing samples with box plots. *Nature Methods*, 11(2), 119–120. <https://doi.org/10.1038/nmeth.2813>)

Thanks for pointing out this glitch in our writing. Since all the box plots were measured from 5 data points, we corrected the figure legends accordingly.

C) T-test on frequencies (proportions) is highly debatable. I would strongly advice against it as frequencies are not normal distributed. While it seems unfortunately almost the standard in biology a population size of three is hardly sufficient for a t-test as well

Indeed, we did not do a t-test on frequencies (previous Figure 4h now new Fig. 6e; p. 59). The error bar, in this case, reflects the SD of mean length based on the distribution of all peptides in the peptide pool. In all cases where we performed a t-test, we included 5 samples.

Minor issues:

1) The writing of the manuscript has a few issues especially regarding grammar. For instance, articles are rarely used. To name one example, it should be degradation by “the”26S proteasome and not just by 26S proteasome. I would recommend letting the manuscript being edited by a native English speaker to avoid grammar mistakes. Also paragraphs contain casual language that should be made more precise.

We corrected accordingly. Thank you for alerting the authors. We ran the text for grammatical errors and fixed the articles according to the advice we received. We also tried to eliminate non-scientific language. In the revised file we have systematically used either “The 20S proteasome” or “26S proteasomes” throughout (can be seen in the track changes comparison file attached).

2) Cryo EM does not produce an “electron density”. So please don’t refer to it (p.8)

We excised any reference to electron density in the context of CryoEM maps.

p. 11: “Loss of density in the cis α -ring”... “Comparing density maps of 20S complexes”

p. 22: “loss of density in the center of the 20S α -ring”

p. 55 "... The difference map (red density) was generated by using a 180° rotated (around the pseudo-2-fold axis) map minus its original unrotated map (transparent grey). **c**, The top view of density maps from the cis α -ring of 20S only (Blue) and the S1 class of 20S+monoUb CyclinB1-NT (brick red) dataset. **d**, Comparison of the 20S alone (transparent blue) and the S1 class of 20S+MonoUb-CyclinB1-NT (brick red) maps in the two β -rings focused on the β -annulus regions. Note the density loss in some of the annulus loops..."

3) The statement that the purified 26S proteasomes don't contain associated DUBs is slightly misleading as they of course must contain rpn11 which is an intrinsic DUB. Also Extended data Figure only checks for the presence of USP14 but there are more proteasome associated DUBs known.

In writing, we tried to differentiate between the intrinsic DUB (Rpn11) and other proteasome-associated DUBs (USP14, UCHL5) that are not found in all proteasome preparations. In order to decrease potential sources of confusion, we clarified in the revised text that these proteasome preparations lack transiently associated DUBs. In view of this comment, we analysed the composition of 20S by MS/MS and note in new Supplementary Fig. 1 that we did not detect any proteasome-associated DUBS.

p. 6: "26S proteasomes (lacking transiently associated DUBs)"

Supplementary file p. 6 "We noted that neither USP14 nor UCHL5 were detected in the 20S sample by MS/MS."

4) The presence of supplementary and extended data Figures at the same time is very confusing and there should be only one kind of supplementary material.

We have now arranged the figures and supplementary material according to Nature Communications guidelines.

5) The paper contains mixed nomenclature for proteasome subunits. e.g. the name PSMD2 is used alongside Rpn10. As the proteasome nomenclature is confusing enough I would suggest to stick either to the PSM or the Rpn nomenclature and don't mix both.

In the revised manuscript, we chose the PSM (human) nomenclature throughout.

For instance:

p. 3: "To do so, the 19S RP utilizes three ubiquitin receptors (PSMD2/Rpn1, PSMD4/Rpn10, "

p. 12: "by knocking down the gene for the essential 19S RP subunit – PSMD2/Rpn1"

p. 60: "The 19S RP subunit PSMD1, and 20S CP subunits PSMA1"

6) Ext. Fig. 2: Figure legend is wrong explained. Both substrates are in a and b. But the difference between a and b is the 20S or the 26S proteasome.

Indeed, thank you. We corrected the figure legend in the new Supplementary Fig. 2a & b.

7) Sup. Fig. 1b: One grey dot has more than 2-fold preference for the 26S proteasome (between P1=30-40).

Indeed. We corrected the dot color (now Supplementary Fig.3b).

8) Fig. 3a: The Bortezomib control seems somewhat random and half-hearted. If used it should be used in both circumstances.

As per the request, we added the lane with Bortezomib-treated 26S to Figure 3a.

9) Fig. 3a,b: bands for tetra Ub and tetUb-CycB1 in a have about 10 kDa more than in b.

The gel in panel A is a Tris-tricine gel whereas panel B is a Tris-glycine gel. We have also noticed that many proteins migrate differently in different gel types.

10) Fig. 3b: add an anti-HA WB to see whether deubiquitinated CycB1 accumulates.

The anti-HA blot of this reaction is now shown in Figure 2D (Figure 3 focuses on the fate of Ub and comes after the confirmation that CyclinB1 does not accumulate in Figure 2).

11) Fig. 3c: supplement anti-Ub/anti-HA WB.

This information was included in extended figure 2; now new Supplementary Fig. 4b.

12) Sup. Fig. 2: include some detail and rationale about the used inhibitors. (IAN, OPA)

For clarity, we now include some detail and rationale about the inhibitors used in the legend. The information can be found: p.9: "...different inhibitors: MG132 – a 20S proteasome inhibitor; IAN, iodoacetamide – an inhibitor for cysteine-based DUBs; OPA, 1,10(O)-phenanthroline – an inhibitor for Rpn11; Na₃VO₄, sodium ortho-vanadate – an ATPase inhibitor; EDTA – a metal ion chelator (for trace metalloproteases or as an ATPase inhibitor by neutralizing residual MgCl₂); geldanamycin – an HSP90 inhibitor; PMSF – a broad specificity protease inhibitor."

13) Ext. Fig. 2d in text = Ext. Fig. 3d.

Correct. We now refer to the correct figure panel (which has moved to the main text as Fig. 4d).

14) Ext. Fig. 4b: The figure indicates that a "Poly-ubiquitin" AB was used for the WB. Wherefore were HA-CycB1-NT expressing HEK293T cells used, when the entirety of ubiquitinated proteins are monitored?

The purpose of ext. Fig 4b (now Fig. 5b) is to demonstrate that HA-CycB1-NT is degraded by proteasome but not by lysosome (top panel; WB anti-cyclin B1). The middle panel (anti-ubiquitin) is a control that shows that bortezomib was effective in stabilizing polyubiquitin conjugates in this cell line.

15) Ext. Fig. 5a: show the whole blot. The CycB1 band could decrease due to ubiquitination.

As per NPG guidelines, we are uploading whole blots of all panels as "Source Data". Specifically, in this panel, we do not see evidence for accumulation of high MW forms of cyclin B1 that could indicate polyubiquitination. Furthermore, in the left lanes, ubiquitination is blocked by E1 inhibitor (PYR-41). We note that Ext. Fig. 5a is now new Supplementary Fig. 10a.

16) Ext. Fig. 5g: show the whole blot. For WT CycB1, only non-ubiquitinated band is shown.

As per NPG guidelines, we are uploading whole blots of all panels as "Source Data". Specifically, in this panel, we do not see evidence for accumulation of high MW forms of cyclin B1 that could indicate polyubiquitination.

17) Ext. Fig. 5h: Why is K0 CycB1 present as multiple bands?

In order to address this astute observation, we are including here a blot testing the anti-CyclinB1 antibody on control HEK293 extracts and HEK293 cells expressing WT-cyclinB1 or K0-CyclinB1 (Fig R4). To summarize: the 55 KDa band reflects endogenous cyclin B1, the ~38 KDa band reflects the NS3-cyclinB1-NT precursor. The ~35 KDa band (?) may reflect a cleaved endo-CyclinB1 fragment or a non-specific band. The ~18 KDa is a monoubiquitinated modified cyclin B (as confirmed by the right-hand panel of immunoprecipitated Cyclin B1 in Ext. Fig. 5h/new Fig. 5h in the revised manuscript) and importantly is detected only for WT cyclin B1 and not for the K0-Cyclin B1. The trace amount of Cyclin B1 that runs just above the HA-cyclin B1NT (??) is most likely a non-specific band.

Fig. R4: Cell lysates from control HEK293, and HA-CyclinB1-NT or K0-CyclinB1-NT transfected HEK293 cells were separated by SDS-PAGE and IB was performed using anti-CyclinB1-NT antibody.

18) Fig. 4b,g: n=3, which is too low for a box plot and questionable for statistical tests.

Thanks for pointing out this glitch in our writing. Since the box plot in Fig. 4b (now new Fig. 6a) was calculated from 5 data points, we corrected the figure legend accordingly. Fig. 4g (now new Fig. 6d) includes more than 100 data points reflecting all the peptides identified in multiple data sets.

19) Fig. 4b,d: Data derives from immunoblot intensities.

Correct. Indicated in the figure legend of new Fig. 6a and Fig. 8a.

20) Fig. 4c: Only non-ubiquitylated CycB1 is shown. Show the whole blot.

As per NPG guidelines, we are uploading whole blots of all panels as “Source Data”.

21) Ext. Fig. 6d: Show the whole blot to see if poly-Ub-alpha-Syn accumulates.

In the revised figure (new Supplementary Fig. 11d), we expanded the cropped area to show a wide range of MW demonstrating that no poly-Ub-alpha-Syn was detected in this experiment.

22) Ext. Fig. 7b: Data derive from WB band quantification.

Correct. Indicated in the figure legend of the new Fig. 7b.

23) Fig. 4e: Show the whole blot to see ubiquitinated CycB1.

As per NPG guidelines, we are uploading whole blots of all panels as “Source Data”.

24) Sup. Fig. 6a: “Pure 26S” contains some PA200 and “Pure 20S” contains some PA28alpha. Supplement PA28beta/gamma IB.

We analysed the purified 20S by MS/MS and identified trace amounts of PA28alpha and beta, but no detectable amounts of gamma (new Supplementary Fig. 14a).

25) Fig 4g: Which test was used? T-test probably not valid here.

We thank the reviewer for pointing this out. We replotted the data points showing the size range and average size length (new Fig. 6d).

26) Fig. 4h: The data do not really seem significant.

In the manuscript, we are careful not to refer to the difference as significant. Nevertheless, we do find it exciting that the comparison of the entire intracellular peptidome in two cell lines is compatible with the proposed role of the 20S in cells. Fig. 4h is now new Fig. 6e.

27) Sup. Fig. 6b,c,d in text = Sup. Fig. 7b,c,d.

We have corrected the figure numbering; these panels are now Supplementary Fig. 12b and Fig.6. b,c.

28) Fig. 5a: show 0h CHX chase of Hi20S.

We have added a zero time point to the CHX chase condition in a new figure in the revised manuscript (Supplementary Fig. 13b).

29) Ext. Fig. 7b in text: The figure shows increased abundance of 20S during hypoxia, not enhanced function.

We corrected the text as suggested by the reviewer. This information can be found: p. 15: “While documenting the proteasome content in cultured cell lines we found that HeLa cells growing under hypoxia (1% O₂) showed a time-dependent shift from a majority of 30S/26S proteasomes to primarily 20S species (Fig. 7a,b).”

30) Fig. 5c: Data derive from WB => No linear relationship.

While estimating protein levels by immunoblotting should be taken extremely cautiously, in this case, we benefited from the fact that both 20S and 26S complexes contain the same 20S subunits at a 1:1 ratio. We clarify that we estimated the ratio only for complexes run in the same gel (native), using the same antibody (PSMA1), and at the same exposure.

We take this opportunity to thank the reviewer for thoroughly reading our manuscript and providing extremely useful professional advice.

Reviewer #2 (Remarks to the Author):

The paper of Sahu et al. describes a new and astonishing feature of the 20S proteasome – the degradation of ubiquitinated, unfolded proteins.

It is for some years accepted by now, that the 20S proteasome has a distinctive function in eukaryotic cells, especially under stress conditions. However, whether unfolded, ubiquitinated proteins are degraded by the 20S or the 26/30S form of the proteasome was unknown. Therefore, this manuscript shows that both proteasomal forms contribute and that under proteotoxic stress condition the 20S proteasome is the most active form, as was shown earlier for oxidative stress conditions. However, the finding that the 20S is also able to degrade ubiquitinated proteins is new.

The experiments are well designed and performed. However, a few additional questions should be answered:

1. A bit puzzling is the finding that the 20S is not able to degrade Ub or the Ub-tetramer in its free form. Is the Ub-tetramer not properly folded if attached to the CyclinB1-NT? Or is the CyclinB1-NT just opening the gate? Is a proteasome with an open gate able to degrade Ub and the Ub-tetramer?

We tested both chemically synthesized and enzymatically synthesized free ubiquitin and polyubiquitin chains with 20S proteasome and did not find evidence for proteolysis (new Supplementary Fig. 5). Our best explanation is that ubiquitin itself is not a targeting signal for the 20S. Without an unfolded stretch that initiates engagement, most globular domains (such as ubiquitin) would not be efficiently processed by free 20S complexes. In a previous study, we had tested the premise that ubiquitin structure could be destabilized upon conjugation, yet we did not find any evidence for that (Singh, S.K. et al. Synthetic Uncleavable Ubiquitinated Proteins Dissect Proteasome Deubiquitination and Degradation, and Highlight Distinctive Fate of Tetraubiquitin. *J Am Chem Soc* **138**, 16004-16015 (2016)). In the current study, we propose that an unstructured polypeptide, such as CyclinB1-NT, could associate with the 20S alpha ring, to promote gate opening and once engaged, commits the conjugated ubiquitin tag for proteolysis. We also discussed this idea in a recent review about possible 20S mechanisms (Sahu, I., and Glickman, M. H. 2021, Structural Insights into Substrate Recognition and Processing by the 20S Proteasome. *Biomolecules*. 11). Nevertheless, it is possible that a 20S proteasome with an open gate may degrade Ub or the Ub-tetramer but its open gate conformation should be initiated by a proper signal prior to Ub entry. We noticed that even the 26S proteasome can degrade a tetraUb conjugate only when attached with the substrate by a DUB-uncleavable bond (*J Am Chem Soc*, 2016 **138**, 16004-16015).

2. The authors analyzed the peptide products of the 20S degradation. How does the peptide with the Lys-Ub-linkage look like? Is there any consistent distance between the last peptide bond cleaved and the linkage (N- and C- terminally and in the Ub.chain). Here the products should be characterized in detail. The same is true for the degradation within the Ub-tetramer.

Following this request, we performed new experiments designed to identify these branched peptides. In order to enrich these branched peptides, we conducted a new set of experiments terminating reactions at later time points, thereby enabling most of the substrate to be proteolyzed by 20S. Positive identification of linkages required a strategy (for MS/MS) that can deal with three degrees of freedom in “non-tryptic branched peptides”. We successfully adapted MSFragger for this purpose, and **we are thrilled to obtain positive evidence of branched peptides generated by 20S proteolysis of an isopeptide-linked ubiquitin-conjugate**. A full description of this strategy is included in the revised manuscript with new

figure panels that illustrate the complexity of products around this isopeptide linkage (new Fig. 3d,e). This information can be found in the revised text –

p. 2: “Ubiquitin remnants on branched peptide-products identified by MS/MS and flexibility in the 20S gate observed by Cryo-EM reflect the ability of the 20S to proteolyse an isopeptide-linked ubiquitin-conjugate.”

p. 10: “. In order to address the possibility of a branched peptide entering into the proteolytic chamber, we performed deep analysis of all products by adapting the MSFragger tool to search for variable ubiquitin remnants on proteasome-digested cyclinB1 (Supplementary Data 6). A total of 18 different branched peptide sequences on lysine64 of cyclinB1 were identified with high confidence based on their MS/MS fragmentation spectra (Fig. 3d,e and Supplementary Table 1). This result provides direct evidence for the ability of the 20S proteasome to process a branched polypeptide, specifically in this case to cleave around an isopeptide bond.”

p. 11: “This partially “open gate” conformation is reminiscent of the activated form of the 20S CP¹⁴, and is of sufficient dimensions to enable entry of the branched polypeptide at the site of ubiquitin conjugation to CyclinB1-NT.”

p. 54: “The illustration represents the branched peptides at the isopeptide linkage of the C-terminus of ubiquitin to the K64 residue of cyclinB1 identified by non-tryptic MS/MS of reaction products. Three major types of branched peptides at K64 of CyclinB1 were detected with variants of ubiquitin remnants: K64—GG, K64—GGR, and K64—GGRL. e, MS/MS spectra of one of the detected branched peptides.”

Regarding the distribution of cleavage sites in a polypeptide, we did not observe any consistency between the peptide bond cleaved at the C-terminus of ubiquitin and either N- or C-termini of the attached cyclin remnant. We could identify a wide variety of cyclin peptides around the isopeptide bond with either G, GG, RGG, or even LRGG (from ubiquitin C-terminus) still attached. The full list of cyclin peptides with or without ubiquitin remnant is included in supplementary Datasheet S6.

Reviewer #3 (Remarks to the Author):

In their manuscript „20S as a stand-alone proteasome in cells can degrade the ubiquitin-tag” Sahu et al. investigate how the 20S catalytic complex differs in its function when it is free or part of the intact 26S proteasome.

The authors use state of the art chemical synthesis to generate a set of site-specifically ubiquitylated proteins that can be used as substrates for both 20S and 26S complexes in order to specifically probe their respective roles in protein degradation in vitro. By combining their approach with various mass-spectrometry based readouts the authors are able to dissect specific roles for both 20S and 26S complexes in substrate selection and peptide generation and demonstrate a novel feature of the 20S complex; its ability to degrade the ubiquitin tag along with the target protein.

While it has been shown before that the free 20S complex exerts specific catalytic functions, this clever use of chemical synthesis puts the authors into the unique position to assign function to specific ubiquitylation states of substrate proteins, which clearly both broadens and deepens our understanding of 20S catalytic function.

The authors then go on and show a potential cellular role of the 20S complex in hypoxia and human failing-heart cells and suggest that elevated roles of 20S heighten cell survival under these conditions, which are both of great interest.

Thus, this manuscript provides an extremely timely and valuable resource for those who study

proteasome-mediated protein degradation and should make for a great fit for Nature Communications.

Thank you.

However, some points need clarification.

While the various MS-based workflows appear to be generally solid and executed carefully, some explanation on how intracellular peptidomics was carried out is missing. The experimental side is somewhat explained in the methods section and the capture to Figure S7, but the MS and downstream analysis side needs additional explanation.

We tried to elaborate in the revised version both in results and methods sections, within constraints of clarity and space.

p. 7: “In order to minimize product reprocessing, in this reaction, great excess of the substrate over the enzyme was maintained and proteolysis was terminated early making sure that at no time point more than 40% of the initial substrate was degraded (Fig. 2g). After quenching the reaction, the peptide products were separated from other proteins by ultra-filtration under denaturation conditions and subjected to non-tryptic proteomic analysis. MS/MS analysis of the isolated peptide products showed differential cleavage patterns and peptide distribution for CyclinB1 with or without ubiquitin attached (Fig. 2h,)”

p. 17: “we developed an approach to identify Proteasome-Trapped Peptides (PTPs) by rapidly isolating intact proteasome complexes directly from tissue and extracting peptides for MS/MS sequencing (Supplementary Fig. 16a,b)”.

p. 30: “Proteasome enrichment and Proteasome Trapped Peptides (PTPs) isolation. The native proteasome lysates were subjected to ultra-centrifugation at 100,000 g for 45 min at 4°C. The first ribosome pellet was discarded, and the supernatant was further centrifuged at 150,000g for 4hr at 4°C. Then the pellet was collected, washed and resuspended with ATP buffer. For PTPs isolation 80°C hot water was added to the enriched proteasome suspension, incubated at 80°C for 20 min and processed as described in “Intracellular peptide isolation” section.

Supplementary Fig. 16 p. 27-28 of the supplementary file.

It’s too long to paste here, but pages 34 to 38 detail the methods for C-MS/MS and Mass-spectrometry analysis, Mass-spectrometry data analysis, Bioinformatics analysis of MS/MS data and Statistics.

Regarding the specific degradation of ubiquitylated samples by 20S/26S (extended Data Fig. 2). The data shows that the endpoints of the enzymatic digestions of both mono and di-ubiquitylated CyclinB1 are identical and only slightly differ in their respective rates; moreover, the rates for mono and di-ubiquitylated cyclin B1 appear to be inverted. It would be great if the authors could elaborate on why they think this is the case.

Indeed, the reviewer’s observation is correct: the rate of mono and di-ubiquitylated CyclinB1 proteolysis differ only slightly and are inverted for the 20S and 26S proteasomes (now in Fig. 2 and Supplementary Fig. 2). The trend becomes apparent when looking at the entire panel of substrates (naked, mono, di, tetra). We propose that the addition of ubiquitin units accelerates degradation by 26S due to tighter binding, which

is not the case for the 20S that lacks ubiquitin receptors. Instead, additional ubiquitin impedes degradation since it has to be unraveled stepwise and proteolyzed by 20S.

p. 7: “By binding to ubiquitin receptors, ubiquitin units facilitate the degradation of a tagged substrate by the 26S proteasome, which is not the case for the 20S species that lacks ubiquitin receptors.”... “The ubiquitinated substrate associates to the 20S proteasome via the unstructured stretch but its proteolysis is attenuated on account of the globular ubiquitin domain. Overall, rapid proteolysis of a ubiquitinated unstructured protein is characteristic of 26S proteasomes, whereas rapid proteolysis of a non-ubiquitinated form of the same protein appears to be a signature of the 20S proteasome species”

Fig. 5: 20S proteasome contributes to degradation of ubiquitin conjugates in vivo:
How exactly was the number of proteasomes determined? More importantly, how the number of ubiquitin-derived peptides? Maybe the authors could clarify this.

We didn't calculate the number of proteasome molecules in vivo. Rather the relative ratio of the 20S to 26S was estimated as shown in supplementary figure 8 (now Supplementary Fig. 15). The number of ubiquitin-derived peptides is a direct reflection of the total MS/MS count, also introduced in supplementary Fig. 15.

The authors make some suggestions on why proteolysis of the ubiquitin tag may be a feature of the 20S complex. Could the same goals not be reached without digestion of the tag? -perhaps the authors could speculate a bit further.

We do not think that degradation of ubiquitin is the main function of the 20S, rather it is collateral damage in specific conditions when 26S is depleted. Lacking deubiquitinase ability, the 20S is not guaranteed (nor designed) to release ubiquitin, but apparently can degrade some along with the conjugates. While we do not speculate on this in the text, reducing the ubiquitin pool may afford some benefit to cells under conditions (less 26S) when it is not needed.

Reviewer #4 (Remarks to the Author):

This work by Sahu et al. is a tour-de-force investigation of the novel function of 20S proteasome. Although the role of 26S proteasome in protein degradation has been well established, the potential role of the 20S proteasome was not clear. This study addressed this intriguing question through reconstituted proteasome system combined with well-defined, synthesized polyUb cyclin B1 (N-terminal disordered region, CyclinB1-NT). The results were further corroborated using cultured cells and human heart myocardial tissues. The results obtained revealed interesting distinctions between the 26S and 20S proteasomes in degrading unmodified and ubiquitinated proteins, particularly disordered proteins as represented by CyclinB1-NT. This study is of high significance and suitable for publication in Nature Communications. Below are questions that need to be addressed to further improve an otherwise already strong manuscript.

The 26S proteasome prep used in this study lacked the proteasome-associated DUBs. This affects the protein degradation outcome in terms of rate and peptide distribution. Can the authors do the same

experiment comparing a 26S proteasome with DUBs associated? This is an important experiment for comparison with the degradation study reported in cells in which DUBs are associated with proteasomes.

Indeed, this would be an interesting and insightful experiment. However, as there are multiple DUBs (at least 3, the published ones) associated with proteasomes, proper execution would be very elaborate and beyond the scope of the current study that focuses on the 20S that lacks DUBs. We have touched upon the role of proteasome-associated DUBs in 26S proteolytic efficiency in a previous study (Singh, S.K. et al. Synthetic Uncleavable Ubiquitinated Proteins Dissect Proteasome Deubiquitination and Degradation, and Highlight Distinctive Fate of Tetraubiquitin. *J Am Chem Soc* **138**, 16004-16015 (2016)), and we wish to expand on this in an upcoming manuscript under preparation.

In Fig. 2h there are clear gaps for mono- and tetraUb-CyclinB1-NT, but not in unmodified CyclinB1. What are these regions and why such a difference exists? Please add residue numbers to the graph.

From this biochemical experiment, it appears that 20S is able to cleave at more sites within the unstructured sequence of cyclinB1 as compared to the same sequence as when attached to ubiquitin. At this stage, we cannot say with certainty whether the gaps in modified cyclin are due to inaccessibility (for instance the ubiquitin anchor stalls the substrate at certain orientations) or conversely due to excessive cleave around these same spots that generate peptide products too small to detect by mass spec. Regarding the graph in Fig 2h, the main purpose of the figure was to show the consistencies of cleavage patterns of the three different substrates by the two enzymes differed. Therefore, an unbiased MDS analysis was performed on cleavage sites from all experiments and the resulting dendrogram (figure 2h; left) shows that the cleavage pattern is consistent between enzymes and substrates. In this analysis each grey bar/blue box (figure 2h; top) represents a peptide product positioned based on either its C-terminus or its N-terminal residue along the primary sequence of CyclinB1-NT, therefore, the amino acid residue sequence has not been denoted. However, for your reference here we are adding below (Fig R5) another heatmap designed based on the P1 sites (i.e. representing the N-terminus of each identified peptide). If you feel that this presentation is clearer we could replace the panel in the final version of the manuscript)

Fig. R5: Heat map of CyclinB1-NT-derived cleavage sites for each substrate by 20S or 26S proteasomes (triplicate or duplicate experiments). Each P1 site positioned along the primary sequence of CyclinB1-NT. Color intensity reflects PSM counts for each peptide normalized to the maximum observed counts. The bar plot above (in grey) represents relative sizes of the corresponding maxima. The dendrogram to the left was obtained by performing MDS analysis on the PSM counts of the top 100 peptides and clustering the samples based on the corresponding MDS distances.

Fig. 3d, although Ub peptides were detected by MS/MS, they cannot be attributed specifically to one of the four Ubs except the Ub peptides containing the tag sequences. The figure is misleading and needs to be modified. In Fig. 3c, the mono and diUb-CyclinB1-NT band should be indicated.

Following this comment, we completely revised the way that the product sequences are presented:

P. 53: Fig. 3c.

p. 54: “The illustration represents the ubiquitin peptides obtained from TetraUb-CyclinB1-NT proteolyzed by the 20S proteasome detected by MS/MS analysis overlaid on the primary sequences of the ubiquitin units in the chain. Red colored residues represent the Flag-tag sequence on the distal ubiquitin unit, while green represents the Myc-tag sequence on the proximal ubiquitin[”

The cellular degradation study used CyclinB1-NT fused to NS3 protease as stated in the text. The gel analysis results shown in Fig. 4 seems to suggest that the CyclinB1-NT was detected as a stand-alone protein. Was this due to cleavage of NS3 from the fusion protein? Please clarify. This is important for the correct interpretation of the results.

Yes. This was precisely the strategy. After the expression of the fusion protein NS3-HA-CyclinB1-NT, the NS3 protease auto cleave itself at the C-term of HA-CyclinB1-NT and release the stand-alone protein HA-CyclinB1-NT. The released NS3 domain serves as an expression/loading control.

The degradation of tetraUb-CyclinB1-NT by 20S was much slower (~ 10 fold, with a half-life of 200 mins) than unmodified CyclinB1-NT as shown in the in vitro system. Coupled with the cryo-EM results presented in the manuscript it is reasonable to suggest that the ubiquitinated substrate is bound to the 20S proteasome normally but its proteolysis is much slowed down. Alternatively, the slower processing of polyUb-CyclinB1-NT may be due to less efficient recruitment of the protein to the 20S proteasome. Have these possibilities been addressed? The authors may do a 20S proteasome assay with an equal mixture of unmodified CyclinB1-NT, mono-, di- and tetraUb CyclinB1-NT and see how they are degraded. If the latter is true unmodified CyclinB1-NT is likely degraded first. Otherwise, an inhibition may be observed due to the presence of polyUb-CyclinB1-NT.

This is a great idea raised by the reviewer. Indeed, if a ubiquitinated substrate associate⁷ with the 20S but is proteolyzed slower than a similar unmodified substrate, it should serve as a competitive substrate and would de facto inhibit proteolysis of the latter. To test this insightful suggestion, we compared the rate of cyclin degradation by purified 20S proteasome alone, and in presence of equimolar ubiquitin-cyclin. **As**

we now show in new supplementary Fig. 2c, a marked slowdown of cyclin proteolysis is observed in presence of ubiquitin-cyclin. As the reviewer proposes, this result rules out the notion that ubiquitination decreases recruitment of the substrate to 20S and supports the interpretation that the ubiquitinated substrate binds to the 20S proteasome (via the unstructured stretch) but its proteolysis is attenuated. We made a note of this in the results section relating to Fig. 2.

p. 7: “In order to evaluate whether ubiquitinated proteins are recruited to 20S proteasomes as efficiently as the equivalent unmodified protein, we performed a competition assay: we compared the proteolysis of CyclinB1-NT by the 20S proteasome alone and in the presence of equimolar monoUb-CyclinB1-NT. A marked slowdown of cyclin proteolysis was observed when monoUb-CyclinB1-NT was added to the reaction mixture (Supplementary Fig. 2c). This result suggests that the ubiquitinated protein serves as a competitive substrate for the 20S proteasome and indirectly suggests that both substrates are comparably recruited on the 20S proteasome (Supplementary Fig. 2c).”

Would elevated level of 20S present an issue given that many fully or partially disordered proteins exist in human cells? How is the 20S proteasome’s activity regulated to avoid toxic effect to cells? This should be considered in the discussion.

We added a few sentences in the discussion. Briefly, we would like to raise the idea that the current study reveals a basal activity of the 20S in cells. A well-documented – yet usually overlooked – observation is that all cells contain free 20S (in the current study estimated at around 40%). Rather than being toxic, the current study explains that these 20S relieve proteotoxicity cause by defective proteins or damaged polypeptides. Usually, most IDPs are protected by chaperones or binding partners if they are to survive in the cellular milieu, but those that are not may be removed by a number of mechanisms, including proteolysis by 20S proteasomes.

This part is now discussed in the main text –

p. 22: “The ubiquitous and rather abundant 20S core particle alongside 26S proteasome holoenzymes in most cell types (in the current study estimated at around 40%) has long baffled researchers. With the understanding that the 20S proteasome is a functional enzyme, its presence in cells does not represent merely unassembled proteasome intermediates or released core particles from disassembled holoenzymes, but rather provides a gain-of-function. As demonstrated in the current study, elevated 20S proteasome levels correlate with efficient removal of damaged proteins, providing benefits during hypoxia or stress such as that imposed by the synthesis of truncated nascent polypeptides. Long-term stress in the form of prolonged hypoxia for example, or in samples of human failing heart tissue, may result in a higher 20S-to-26S proteasome ratio to unleash the signature activities of free 20S enzymes. While the degradation of ubiquitin-conjugates is essential for cell viability, the necessity to rapidly relieve the proteotoxicity caused by defective proteins or damaged polypeptides may come at the expense of the small portion of conjugated ubiquitin that is degraded along with the target substrates by 20S proteasome activity. Elevated levels of 20S complexes often thought to reflect a breakdown of proteasome holoenzymes is characteristic of aging cells, yet its presence may be more than merely a marker for aging as 20S proteasomes may also contribute to survival during the aging process accompanied with diminished protein quality control systems.”

REVIEWER COMMENTS

Reviewer #1 (Remarks to the Author):

The comments regarding the scientific content of this manuscript, which have arisen from the initial submission, have been addressed and improved. The main issue with the manuscript - the use of language and overall styling – remains partially unresolved. The overall findings shown in this paper are supported by the data included, but their overall presentation is not at a publishable level yet. However the detailed model presented at the very end is highly speculative and needs to be presented as a hypothesis, with more caution.

Major points:

1) The readability of the manuscript remains the main issue with the manuscript. While this has improved a little, it is still not reader friendly. I would strongly suggest work on the language and the presentation of the manuscript. It is immensely dense and still very hard to follow.

The manuscript still contains multiple grammatical errors. In particular, these include wrong word order in sentences or an overall bad use of grammar. Overall, I would suggest to have the manuscript edited by a professional editor or a native speaker for a better flow of the text.

Points to consider when improving the manuscript:

- A more coherent use of tenses, in particular in figure legends and individual paragraphs
- Correction of typos
- Consistency with the use of capital letters (for example, either choose “cryo-EM” or “Cryo-EM”)

2) I would suggest to revisit the representation of the Cryo-EM data. Despite all previous suggestions being implemented in the revised manuscript, some of the data is not well described. This would include:

- I am convinced the two maps show differences but these have to be analysed and described better to give a meaning to them. I would suggest to present and compare the pdb models of the two maps to get an accurate description of the actual differences. According to the methods part models have been built. But they are not presented in the manuscript. As a technical advice the software DeepEMhancer or other map denoising packages enables a more detailed analysis and will facilitate accurate model building.

- A more in-depth description of the two classes obtained from the 20S-MonoUb-CyclinB-NT dataset where class “S1” is described in detail, but class “S2” lacks explanation.
- A better representation of the 3D variability analysis (Supplementary Figure 9). Example on how to present this data can be found in the referenced article (Punjani & Fleet, 2021)

3) The jump from the actual cryoEM data in Figure 4 to the model in Figure 10 is big and it is rather speculative. This can give the very wrong impression especially with such a detailed description in the final model. For example there is simply no density that is assigned for ubiquitin and this everything else is speculative and such detailed model as presented is simply not justified given all the data. Even combined with all the other data this is not justified in presented detail. I would urgently suggest to tone this down.

Minor comments:

- Check referencing, as it appears some text needs to be referenced better
- Check all abbreviations, some are not described upon first mention. For example “CQN”
- Remove text in lines 418-421 as this is a duplicate
- Please correct the order of several figures (for example, Figure 2 shows “d” before “c”, Figure 3 shows “e” before “d”). Whilst this makes sense space-wise, it is confusing for the reader.

References:

Punjani A, Fleet DJ. 3D variability analysis: Resolving continuous flexibility and discrete heterogeneity from single particle cryo-EM. *J Struct Biol.* 2021 Feb 11;213(2):107702. doi: 10.1016/j.jsb.2021.107702.

Reviewer #3 (Remarks to the Author):

The authors have addressed and clarified all my remaining issues and I am happy for the manuscript to be published in Nature Communications.

Reviewer #4 (Remarks to the Author):

The authors have addressed the reviewer's questions satisfactorily.

POINT-BY-POINT TO THE REVIEWER COMMENTS

Reviewer #1 (Remarks to the Author):

The comments regarding the scientific content of this manuscript, which have arisen from the initial submission, have been addressed and improved. The main issue with the manuscript - the use of language and overall styling – remains partially unresolved. The overall findings shown in this paper are supported by the data included, but their overall presentation is not at a publishable level yet. However the detailed model presented at the very end is highly speculative and needs to be presented as a hypothesis, with more caution.

Major points:

1) The readability of the manuscript remains the main issue with the manuscript. While this has improved a little, it is still not reader friendly. I would strongly suggest work on the language and the presentation of the manuscript. It is immensely dense and still very hard to follow.

Ans: We have worked on this aspect/concern and corrected all the necessary sections. All the changes are left as “track changes” and can be viewed in the uploaded manuscript draft.

The manuscript still contains multiple grammatical errors. In particular, these include wrong word order in sentences or an overall bad use of grammar. Overall, I would suggest to have the manuscript edited by a professional editor or a native speaker for a better flow of the text. Points to consider when improving the manuscript:

- A more coherent use of tenses, in particular in figure legends and individual paragraphs
- Correction of typos
- Consistency with the use of capital letters (for example, either choose “cryo-EM” or “Cryo-EM”)

Ans: We have consulted a professional scientific writer (Avital Bareket-Samish, BioInsight) to provide English editing, correct language, typos and grammatical glitches, and to improve the overall readability of the text. All of the changes are highlighted in the revised file as “track changes” for your consideration.

2) I would suggest to revisit the representation of the Cryo-EM data. Despite all previous suggestions being implemented in the revised manuscript, some of the data is not well described. This would include:

- I am convinced the two maps show differences but these have to be analysed and described better to give a meaning to them. I would suggest to present and compare the pdb models of the two maps to get an accurate description of the actual differences. According to the methods part models have been build. But they are not presented in the manuscript. As a technical advice the software DeepEMhancer or other map denoising packages enables a more detailed analysis and will facilitate accurate model building.

Ans: Thanks to the reviewer for the suggestions. We have followed the recommendation to use DeepEMhancer to post-process our final maps of 20S+MonoUb-CyclinB-NT and 20S. These post-processed maps enabled us to fit a model into each map, as can be seen in the enclosed figure (**Fig. R1a**). Although the processed density maps still show distinct variations, the two models largely overlap. Careful observation of the models reveals variations in the gate region (**Fig. R1a**), and in particular to the reverse turn of several α -subunits (located between the N-terminal tail and H0) (**Fig. R1b**). We note that the reverse turns are key factors in 20S proteasome gate opening [1-4]. However, the objective of the study was not to get high-resolution models but rather to uncover structural differences due to the presence of substrate. Since the original experimental density maps already revealed potential conformational or dynamic variations in the gate region of the substrate-affected maps, we prefer to emphasize this point without further post-processing of the experimental cryo-EM map. RELION is a widely used, time-tested software for this purpose, whereas DeepEMhancer is a new software (just on-line published this month) for automatic post-processing of data (using a deep-learning approach to sharpen atomic models). Therefore, we prefer to keep using the original map in our manuscript (**Fig. R1c**).

Fig. R1 (a) Model-map fitting of 20S+monoUb-CyclinB1-NT and 20S only structures. These maps have been processed by DeepEMhancer. (b) Overlapping of the two models (20S+monoUb-CyclinB1-NT in red; 20S only in blue) highlighting changes to the reverse turn regions (shaded yellow). (c) Density maps of 20S only and 20S+monoUb-CyclinB1-NT obtain by RELION showing top view of 20S with local resolution.

- A more in-depth description of the two classes obtained from the 20S-MonoUb-CyclinB-NT dataset where class “S1” is described in detail, but class “S2” lacks explanation.

Ans: We have now added a new panel for S2 as **Supplementary Fig. 7g**, and gave a more in-depth description in main text to now read: “A similar analysis of substrate-incubated 20S complexes resulted in five classes, with one class (S1) comprising 29.1% of the particles in an

asymmetric appearance (Supplementary Fig. 7a-e). 62.9% of the particles appeared symmetrically closed in the two rings, similar to 20S only structure, and could be refined into a map denoted as S2 (Supplementary Fig. 7d, g). S2 may represent the particles not directly or obviously affected by substrate.”

- A better representation of the 3D variability analysis (Supplementary Figure 9). Example on how to present this data can be found in the referenced article (Punjani & Fleet, 2021)

Ans: Thanks for the suggestion and we have added a figure for 3DVA (**Supplementary Fig 9b, d**) using a similar style to the referenced paper [5]. It is also shown here as **Fig. R2** for the convenience of the reviewers and editor. This comparison shows that the Component 1 of 20S+monoUb-CyclinB-NT exhibits dynamics in the gate region without obvious overall shape change; while Component 1 of the 20S only map depicts a squashed motion of the whole complex without obvious local motion in the gate region.

Fig. R2: Results of 3DVA showing different patterns of motions between 20S+monoUb-CyclinB-NT and 20S only structures. (a) Component 1 of 20S+monoUb-CyclinB-NT exhibits dynamics in the gate region (indicated by orange dashed circle) without obvious overall shape change. Here the red and blue maps are the two representative extreme maps. This rendering style is followed. (b) Component 1 of 20S only map depicts a squashed motion of the whole complex. The dashed circles in the first two panels are identical perfect circles. It can be seen that the distances between the edge of structure and the circle are different in the arrow indicated direction (from blue to red map).

3) The jump from the actual cryoEM data in Figure 4 to the model in Figure 10 is big and it is rather speculative. This can give the very wrong impression especially with such a detailed description in the final model. For example there is simply no density that is assigned for ubiquitin and this everything else is speculative and such detailed model as presented is simply not justified given all the data. Even combined with all the other data this is not justified in presented detail. I would urgently suggest to tone this done.

Ans: We have now altered the model figure “**Figure 10**” as per the suggestion. The new model figure summarizes all biochemical, mass spec and cell biology aspects of the study to highlight the distinct roles of 20S proteasome.

Minor comments:

- Check referencing, as it appears some text needs to be referenced better
- Check all abbreviations, some are not described upon first mention. For example “CQN”
- Remove text in lines 418-421 as this is a duplicate
- Please correct the order of several figures (for example, Figure 2 shows “d” before “c”, Figure 3 shows “e” before “d”). Whilst this makes sense space-wise, it is confusing for the reader.

Ans: Most of these minor suggestions were addressed to the best of our ability.

References (for the letter):

1. Whitby, F. G. et al. Structural basis for the activation of 20S proteasomes by 11S regulators. *Nature* 408, 115–120 (2000).
2. Forster, A., Masters, E. I., Whitby, F. G., Robinson, H. & Hill, C. P. The 1.9 Å structure of a proteasome-11S activator complex and implications for proteasome-PAN/PA700 interactions. *Mol. Cell* 18, 589–599 (2005).
3. Forster, A., Whitby, F. G. & Hill, C. P. The pore of activated 20S proteasomes has an ordered 7-fold symmetric conformation. *EMBO J.* 22, 4356–4364 (2003).
4. J, Chen#, Y. Wang#, C. Xu#, K. Chen, Q. Zhao, S. Wang, Y. Yin, C. Peng*, Z. Ding* and Y. Cong* (2021). "Cryo-EM of mammalian PA28 $\alpha\beta$ -iCP immunoproteasome reveals a distinct mechanism of proteasome activation by PA28 $\alpha\beta$." *Nat Comms* 12(1): 739.
5. Punjani A, Fleet DJ. 3D variability analysis: Resolving continuous flexibility and discrete heterogeneity from single particle cryo-EM. *J Struct Biol.* 2021 Feb 11;213(2):107702. doi: 10.1016/j.jsb.2021.107702.

REVIEWER COMMENTS

Reviewer #1 (Remarks to the Author):

I do thank the authors for improving on most of my concerns.

However, one concern remains which is the structural presentation. In my view, if you attempt structural investigation you will need to go till the end and you certainly can. So you should provide interpreted pdb models that clearly show the differences. Otherwise the structural analysis is meaningless. By just showing the density difference you suggest that there is a meaningful difference. In your response to me, you showed that there are some differences but that they are minor however they still might be meaningful in the important region. I don't see any reason not to include a proper analysis in the paper with a model and describe exactly this. This is not even much work as you (at least partially) did it already. Without a pdb model, I believe the structural work is unfinished. In your mass spec work you also assign sequence to your fragments and not just show the peaks. Such figures as you made for me should be included in the manuscript at least in the supplement such that readers can judge.

You suggest "Since the original experimental density maps already revealed potential conformational or dynamic variations in the gate region of the substrate-affected maps, we prefer to emphasize this point without further post-processing of the experimental cryo-EM map." If this is the case please show that in an interpreted model. DeepEMhancer was supposed to help you there. I am fine to not use it if your claims can be supported otherwise.

I am still excited about the paper and think it should be published but it should be done properly.

RESPONSE TO REVIEWER COMMENTS

Reviewer #1 (Remarks to the Author):

I do thank the authors for improving on most of my concerns.

However, one concern remains which is the structural presentation. In my view, if you attempt structural investigation you will need to go till the end and you certainly can. So you should provide interpreted pdb models that clearly show the differences. Otherwise the structural analysis is meaningless. By just showing the density difference you suggest that there is a meaningful difference. In your response to me, you showed that there are some differences but that they are minor however they still might be meaningful in the important region. I don't see any reason not to include a proper analysis in the paper with a model and describe exactly this. This is not even much work as you (at least partially) did it already. Without a pdb model, I believe the structural work is unfinished. In your mass spec work you also assign sequence to your fragments and not just show the peaks. Such figures as you made for me should be included in the manuscript at least in the supplement such that readers can judge.

You suggest "Since the original experimental density maps already revealed potential conformational or dynamic variations in the gate region of the substrate-affected maps, we prefer to emphasize this point without further post-processing of the experimental cryo-EM map." If this is the case please show that in an interpreted model. DeepEMhancer was supposed to help you there. I am fine to not use it if your claims can be supported otherwise.

I am still excited about the paper and think it should be published but it should be done properly.

Ans:

We understand the reviewer's insistence that we exhaust all structural investigation efforts and provide processed, post-processed and modeled data. We note that processed density maps by DeepEMhancer or RELION still show distinct variations between 20S alone and 20S+substrate. Nevertheless, the two generated models largely overlap. To illustrate this point, we include in the current revision a model superimposed on each of the post-processed maps as supplementary Figure 9. Careful observation of the models reveals variations in the gate region (Supplementary Fig. 9a) and in particular to the reverse turn of several α -subunits (Supplementary Fig. 9b). Importantly, we have now completed deposition of maps to EMDB and of the models to PDB and provide the accession numbers in the main text. We hope this provides the reader with all the information and tools needed to critically read our manuscript and take home the message they find pertinent to their interests.